# DISTANCE-BASED BACKGROUND CLASS REGULARIZATION FOR OPEN-SET RECOGNITION

## ABSTRACT

In open-set recognition (OSR), classifiers should be able to reject unknown-class samples while maintaining robust closed-set classification performance. To solve the OSR problem based on pre-trained Softmax classifiers, previous studies investigated offline analyses, e.g., distance-based sample rejection, which can limit the feature space of known-class data items. Since such classifiers are trained solely based on known-class samples, one can use background class regularization (BCR), which employs background-class data as surrogates of unknown-class ones during training phase, to enhance OSR performance. However, previous regularization methods have limited OSR performance, since they categorized known-class data into a single group and then aimed to distinguish them from anomalies. In this paper, we propose a novel distance-based BCR method suitable for OSR, which limits the feature space of known-class data in a class-wise manner and then makes background-class samples located far away from the limited feature space. Instead of conventional Softmax classifiers, we use distance-based classifiers, which utilize the principle of linear discriminant analysis. Based on the distance measure used for classification, we design a novel regularization loss function that can contrast known-class and background-class samples while keeping robust closed-set classification performance. Through our extensive experiments, we show that the proposed method provides robust OSR results with a simple inference process.

## 1 INTRODUCTION

In machine learning (ML), classification algorithms have achieved great success. Through recent advances in convolutional neural networks, their classification performance already surpassed the human-level performance in image classification (He et al., 2015). However, such algorithms are usually developed under a *closed-set* assumption, i.e., the class of each test sample is assumed to always belong to one of the pre-defined set of classes. Although this conventional assumption can be easily violated in real-world applications (classifiers can face unknown-class data), traditional algorithms are likely to force unknown-class samples to be classified into one of the known classes. To tackle this issue, the *open-set recognition (OSR)* problem (Scheirer et al., 2013) aims to properly classify unknown-class samples as "unknown" and known-class samples as one of the known classes.

According to the definition of OSR (Scheirer et al., 2013), it is required to properly limit the feature space of known-class data. To satisfy the requirement, various OSR methods were developed based on traditional ML models. Previously, Scheirer et al. (2014) calibrated the decision scores of support vector machines (SVMs). Based on the intuition that a large set of data samples of unknown classes can be rejected if those of known classes are accurately modeled, Jain et al. (2014) proposed $P_I$-SVM, which utilized the statistical modeling of known-class samples located near the decision boundary of SVMs. Afterward, Júnior et al. (2016) attempted to solve the OSR problem based on the principle of the nearest neighbors. Taking distribution information of data into account, Rudd et al. (2018) proposed the extreme value machine by utilizing the concept of margin distributions.

Since deep neural networks (DNNs) have robust classification performance by learning high-level representations of data, OSR methods for DNNs have received great attention. Based on the theoretical foundations used in traditional ML-based OSR methods, Bendale & Boult (2016) proposed the first OSR strategy for DNNs called Openmax, which calibrates the output logits of pre-trained Softmax classifiers. To improve Openmax, Yoshihashi et al. (2019) proposed the classification-reconstruction

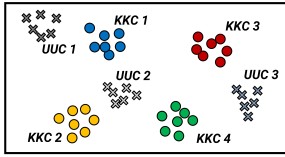 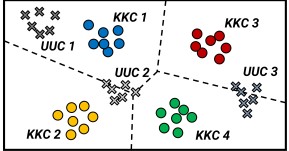 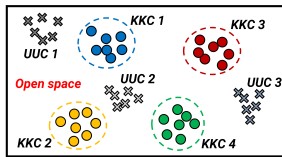

(a) Latent feature space     (b) Closed-set classification     (c) Open-set classification

Figure 1: Given (a) a latent feature space, we demonstrate (b) closed-set and (c) open-set classification examples, where KKCs and UUCs are known and unknown classes, respectively.

learning to make robust latent feature vectors. Afterward, Oza & Patel (2019) proposed to exploit a class-conditioned autoencoder and use its reconstruction error to assess each input sample. Sun et al. (2020) employed several class-conditioned variational auto-encoders for generative modeling.

Although previous OSR methods using discriminative models applied *offline analyses* to pre-trained Softmax classifiers or employed complicated DNN models, they have limited performance since the classifiers were trained solely based on known-class samples. To mitigate the problem, one can use background class regularization (BCR) to achieve robust empirical results. However, previous BCR methods (Dhamija et al., 2018; Hendrycks et al., 2019; Liu et al., 2020) are insufficient to properly solve the OSR problem. To design an effective open-set classifier that can overcome the previous limitations, we propose a novel BCR method suitable for OSR, which uses a distance-based classifier and a novel loss function for regularization. We provide detailed description in the next section.

## 2 PRELIMINARIES AND OUR CONTRIBUTIONS

### 2.1 THE OPEN-SET RECOGNITION PROBLEM

The OSR problem addresses the classification setting that can face test samples from classes unseen during training. In this setting, open-set classifiers aim to properly classify *known-class* samples while rejecting *unknown-class* samples simultaneously. A similar problem to OSR is out-of-distribution (OOD) detection (Hendrycks & Gimpel, 2017), which typically aims to reject data items drawn far away from the training data distribution. In previous OOD detection studies (Hendrycks & Gimpel, 2017; Liang et al., 2018; Lee et al., 2018a;b), OOD samples tend to be drawn from other datasets or even images of noise. In this paper, we aim to reject test data whose classes are unknown but related to the training data, which narrows down the scope of conventional OOD detection tasks.

Previously, Scheirer et al. (2013) introduced a formal definition of the OSR problem based on the notion of open-space risk $R_{\mathcal{O}}$, which is a relative measure of a positively labeled union of balls $\mathcal{S}_V$ and open space $\mathcal{O}$ located far from $\mathcal{S}_V$. Since labeling any data item in $\mathcal{O}$ incurs open-space risk, it is straightforward that a classifier cannot be a solution for the OSR problem if the classifier accepts data in infinitely wide regions, i.e., its open-space risk is unbounded ($R_{\mathcal{O}} = \infty$). The definition implies that essential requirements to solve the OSR problem are 1) *bounding open-space risk* and 2) *ideally balancing it with empirical risk* to maintain a low classification error rate.

Unlike traditional closed-set classifiers, open-set classifiers are required to limit the space of known-class data to bound their open-space risk. To ensure bounded open-space risk, Scheirer et al. (2014) proposed to formulate compact abating probability (CAP) models. The principle of CAP models is that if the support region of a classifier decays in all directions from the training data, thresholding the region will bound the classifier's open-space risk (Boult et al., 2019). As depicted in Figure 1, which compares traditional closed-set and open-set classification problems, properly building *class-wise* CAP models is an effective strategy to satisfy the two essential requirements of OSR.

### 2.2 POST-CLASSIFICATION ANALYSIS FOR PRE-TRAINED SOFTMAX CLASSIFIER

In this paper, we aim to solve the OSR problem by using a standard DNN-based classifier architecture $f$ as a latent feature extractor. Applying a fully-connected layer to $f$, a conventional Softmax classifier

computes the posterior probability of an input $\mathbf{x}$ belonging to the $c$-th known class by

$$P_s(y = c|\mathbf{x}) = \frac{\exp(\mathbf{w}_c^T f(\mathbf{x}) + b_c)}{\sum_{i=1}^{C} \exp(\mathbf{w}_i^T f(\mathbf{x}) + b_i)}, \tag{1}$$

where $c \in \{1, \cdots, C\}$, $f(\mathbf{x}) \in \mathbb{R}^n$ is the latent feature vector of $\mathbf{x}$, and $\mathbf{w}_c$ and $b_c$ are the weight and the bias for the $c$-th class, respectively. For pre-trained Softmax classifiers, Hendrycks & Gimpel (2017) proposed a baseline to detect anomalous samples, which imposes a threshold on the predictive confidence of Eq. (1). When using the baseline method to solve the OSR problem, one can estimate the class of each known-class sample and recognize unknown-class data by

$$\widehat{y} = \begin{cases} \arg\max_{c \in \{1, \cdots, C\}} P_s(y = c|\mathbf{x}), & \text{if } \max_{c \in \{1, \cdots, C\}} P_s(y = c|\mathbf{x}) \geq \tau, \\ C + 1 \text{ (unknown class)}, & \text{otherwise.} \end{cases} \tag{2}$$

However, Eq. (2) cannot formally bound open-space risk since it only rejects test samples near the decision boundary of classifiers, thereby having infinitely wide regions of acceptance (Boult et al., 2019). Therefore, additional *post-classification analysis* methods using an auxiliary measure other than the Softmax probability are necessary to bound open-space risk, where *distance measures* have been widely used to build auxiliary CAP models in the latent feature space of $f$.

To construct class-wise CAP models, Openmax (Bendale & Boult, 2016) defined radial-basis decaying functions $\{s(\mathbf{x}, i)\}_{i=1}^{C}$, each of which measures the *class-belongingness* of $\mathbf{x}$ for the $c$-th class, in the latent feature space of $f$. For each $s(\mathbf{x}, c)$, the authors employed distance measures between $f(\mathbf{x})$ and an empirical class mean vector $\boldsymbol{\mu}_c$, e.g., $s(\mathbf{x}, c) = D_E^2(f(\mathbf{x}), \boldsymbol{\mu}_c) = (f(\mathbf{x}) - \boldsymbol{\mu}_c)^T (f(\mathbf{x}) - \boldsymbol{\mu}_c)$. To formulate effective CAP models based on $s(\mathbf{x}, c)$, they statistically modeled the distribution of $s(\mathbf{x}, c)$ for known-class data based on the extreme value theory (EVT) (Scheirer, 2017), which provides a theoretical foundation that the Weibull distribution is suitable for modeling known-class samples located far from the class mean vectors (extreme samples). To be specific, Openmax fits a Weibull distribution on extreme samples of the $c$-th class having the highest $D_E(f(\mathbf{x}), \boldsymbol{\mu}_c)$ values, where its cumulative distribution function (CDF) formulates the *probability of inclusion* $P_I(\mathbf{x}, c)$ (Jain et al., 2014; Rudd et al., 2018), i.e., $P_I(\mathbf{x}, c) = 1 - \texttt{WeibullCDF}$, which rapidly decays near extreme samples. Based on $P_I(\mathbf{x}, c)$, the decision rule of Eq. (2) can be calibrated to conduct robust OSR.

## 2.3 Generalized Open-Set Recognition and Background Class Regularization

Although they require additional inference procedures (e.g., EVT modeling) to bound the open-space risk of pre-trained Softmax classifiers, previous post-classification analyses may have limited OSR performance since the classifiers are trained solely based on known-class samples. To empirically obtain robust OSR results without complicated post-classification analyses, one can use the strategy of BCR at the training phase, which exploits background-class samples as surrogates of unknown-class samples. Geng et al. (2020) argued that the *generalized* OSR setting that utilizes background samples is still less-explored and an important research direction to improve OSR performance. In the generalized OSR setting, classifiers should consider the following data classes among the infinite label space of all classes $\mathcal{Y}$ (Dhamija et al., 2018; Geng et al., 2020).

- Known known classes (KKCs; $\mathcal{K} = \{1, \cdots, C\} \subset \mathcal{Y}$) include distinctly labeled positive classes, which also have side information, where $\mathcal{U} = \mathcal{Y} \setminus \mathcal{K}$ is the entire unknown classes.
- Known unknown classes (KUCs; $\mathcal{B} \subset \mathcal{U}$) include background classes, e.g., labeled negative classes which are not necessarily grouped into a pre-defined set of KKCs $\mathcal{K}$.
- Unknown unknown classes (UUCs; $\mathcal{A} = \mathcal{U} \setminus \mathcal{B}$) represent the rest of $\mathcal{U}$, where UUC samples are not available at training time and have no side information, but occur at inference time.

Throughout this paper, we denote the corresponding datasets for the data classes as follows. $\mathcal{D}_t$ is a training set consisting of multiple pairs of a KKC data sample and the corresponding class label $y \in \{1, \cdots, C\}$. $\mathcal{D}_{test}^k$ and $\mathcal{D}_{test}^u$ are test sets of KKCs and UUCs, respectively. $\mathcal{D}_b$ is a background dataset of KUCs. Then, a loss function for training classifiers with BCR can be formulated by

$$\mathcal{L} = \mathcal{L}_{cf} + \lambda \mathcal{L}_{bg} = \mathbb{E}_{(\mathbf{x}^k, y) \sim \mathcal{D}_t} \left[ -\log P_s(y|\mathbf{x}^k) + \lambda \mathbb{E}_{\mathbf{x}^b \sim \mathcal{D}_b} \left[ f_{reg}\left(\mathbf{x}^k, y, \mathbf{x}^b\right) \right] \right], \tag{3}$$

where $\mathcal{L}_{cf}$ and $\mathcal{L}_{bg}$ are the losses for classification and BCR, respectively, and $\lambda$ is a hyperparameter that balances $\mathcal{L}_{cf}$ and $\mathcal{L}_{bg}$. For $\mathcal{L}_{bg}$, previous studies designed their own $f_{reg}$, where Dhamija et al.

(2018) proposed the objectosphere loss for OSR, and Hendrycks et al. (2019) and Liu et al. (2020) employed the uniformity and the energy losses for OOD detection, respectively.

In this paper, we tackle the following limitations of the previous BCR methods.

- In the previous BCR methods, $\mathcal{L}_{bg}$ were designed to make normal data and anomalies more distinguishable in terms of the corresponding anomaly scores. Since they categorized normal samples into a single group (do not consider the classes) in $\mathcal{L}_{bg}$, the previous methods cannot be suitable for rejecting UUC samples located relatively close to KKC data and maintaining robust closed-set classification results.
- The previous methods using the decision rule of Eq. (2) (e.g., objectosphere (Dhamija et al., 2018) and uniformity (Hendrycks et al., 2019)) cannot bound open-space risk. Although one can use post-classification analyses to bound open-space risk, trained latent feature spaces are likely to be inappropriate for using another metric such as distance measures.
- To increase the gap between KKC and KUC samples in terms of latent feature magnitude and energy value in the objectosphere (Dhamija et al., 2018) and the energy (Liu et al., 2020) losses, respectively, it is necessary to find proper margin parameters for each dataset.

### 2.4 Our Contributions

Based on a standard DNN-based classifier architecture $f$, we aim to design open-set classifiers having a simple yet effective inference process by using the principle of BCR. To overcome the limitations described in Section 2.3, we use distance-based classifiers and propose a novel BCR strategy based on the framework of Eq. (3). In the following, we summarize our proposed strategy.

- Instead of applying fully-connected layers to feature extractors $f$, we use the principle of linear discriminant analysis (LDA) (Murphy, 2012) to classify each input based on a distance measure. By imposing a threshold on the distance measure as in Eq. (2), such distance-based classifiers can bound their open-space risk without additional offline analyses.
- To enhance the OSR performance of distance-based classifiers, we propose a novel distance-based BCR strategy. For $\mathcal{L}_{bg}$, we design a loss function that does not require data-dependent margin parameters. Our method first limits the feature space of KKC data by formulating class-wise boundaries based on the concept of the probability of inclusion, and then forces KUC data to be located outside the entire boundaries. By imposing additional regularization on each KKC data item to be located inside the corresponding class-wise boundary, our proposed method can also maintain robust closed-set classification performance.

## 3 Proposed Method

### 3.1 Distance-Based Classification Models

To train a robust open-set classifier as we described in Section 2.4, we use a *distance-based classifier*

$$P_d(y = c | \mathbf{x}) = \frac{P(y = c) \cdot \mathcal{N}(f(\mathbf{x}) | \boldsymbol{\mu}_c, \mathbf{I})}{\sum_{i=1}^{C} P(y = i) \cdot \mathcal{N}(f(\mathbf{x}) | \boldsymbol{\mu}_i, \mathbf{I})} = \frac{P(y = c) \cdot \exp\left(-D_E^2(f(\mathbf{x}), \boldsymbol{\mu}_c)\right)}{\sum_{i=1}^{C} P(y = i) \cdot \exp\left(-D_E^2(f(\mathbf{x}), \boldsymbol{\mu}_i)\right)} \quad (4)$$

as an alternative of Eq. (1), where Eq. (4) uses the principle of LDA in $f(\mathbf{x})$. In Eq. (4), we exploit an identity covariance matrix $\mathbf{I}$ and $P(y = c) = C^{-1}$ for all $c$ for KKC data. Then, Eq. (4) classifies each $\mathbf{x}$ by computing $D_E^2(f(\mathbf{x}), \boldsymbol{\mu}_c) = (f(\mathbf{x}) - \boldsymbol{\mu}_c)^T(f(\mathbf{x}) - \boldsymbol{\mu}_c)$, the Euclidean distance between $f(\mathbf{x})$ and $\boldsymbol{\mu}_c \in \mathbb{R}^n$, where we call $\boldsymbol{\mu}_c$ a *class-wise anchor*. Instead of updating or computing empirical $\boldsymbol{\mu}_c$, we randomly sample each $\boldsymbol{\mu}_c$ from the standard Gaussian distribution for all $c$ and then fix it as an anchor during the training process. Such strategy also showed successful results in Izmailov et al. (2020), which formulated *generative classifiers* based on the principle of Gaussian mixture models.

At inference time, each KKC sample $\mathbf{x}$ can be classified by $\widehat{y} = \arg\min_{c \in \{1, \cdots, C\}} D_E^2(f(\mathbf{x}), \boldsymbol{\mu}_c)$. It is noteworthy that thresholding $D_E^2(f(\mathbf{x}), \boldsymbol{\mu}_c)$, which is the same metric used for classification, can bound the open-space risk of a distance-based classifier by formulating class-wise CAP models. Therefore, one can properly conduct OSR without additional post-classification analyses as follows:

$$\widehat{y} = \begin{cases} \arg\min_{c \in \{1, \cdots, C\}} D_E^2(f(\mathbf{x}), \boldsymbol{\mu}_c), & \text{if } \max_{c \in \{1, \cdots, C\}} -D_E^2(f(\mathbf{x}), \boldsymbol{\mu}_c) \geq \tau, \\ C + 1 \text{ (unknown class)}, & \text{otherwise.} \end{cases} \quad (5)$$

Distance-based classification methods were also employed in prototypical networks (Snell et al., 2017), nearest class mean classifiers (Mensink et al., 2012), and the previous studies of the center loss function (Wen et al., 2019) and convolutional prototype classifiers (Yang et al., 2020). Furthermore, polyhedral conic classifiers (Cevikalp et al., 2021) used the idea of returning compact class regions for KKC samples based on distance-based feature analyses. It is noteworthy that our main contribution is a novel BCR method that can effectively utilize KUC samples in a distance-based classification scheme (described in Sections 3.2 and 3.3), not the distance-based classifier method itself. To the best of our knowledge, we are the first to discuss the necessity of distance-based BCR methods for OSR and propose a reasonable regularization method for distance-based classifiers.

## 3.2 BACKGROUND CLASS REGULARIZATION FOR DISTANCE-BASED CLASSIFIERS

To obtain robust OSR performance based on distance-based classifiers using the simple inference of Eq. (5), we aim to design a BCR method, which uses $\mathcal{D}_t$ and $\mathcal{D}_b$ as surrogates of $\mathcal{D}_{test}^k$ and $\mathcal{D}_{test}^u$ at training time, respectively. Although it cannot provide any information of $\mathcal{D}_{test}^u$, $\mathcal{D}_b$ can be effective to limit the latent feature space of KKC data, while reserving space for UUC samples. Note that the decision rule of Eq. (5) conducts both closed-set classification and unknown-class rejection based on a single distance measure $D_E^2(f(\mathbf{x}), \boldsymbol{\mu}_c)$. Thus, it is intuitive that the primary objective of our BCR method is to make KUC samples located far away from $\boldsymbol{\mu}_i$ for all $i \in \{1, \cdots, C\}$.

Before we illustrate our BCR method, we first introduce hypersphere classifiers (HSCs) (Ruff et al., 2020), whose concept was also used in (Liznerski et al., 2021). An HSC conducts anomaly detection by using a DNN-based feature extractor $g$, where its anomaly score is the Euclidean distance between a center vector $\boldsymbol{\mu}$ and the latent feature vector $g(\mathbf{x})$ of each input $\mathbf{x}$. When training the HSC model, the authors used normal and background data, $\mathcal{D}_t$ and $\mathcal{D}_b$, respectively, and a loss function

$$\mathcal{L}_{hsc} = \mathbb{E}_{\mathbf{x}^k \sim \mathcal{D}_t} \left[ h \left( D_E^2 \left( g(\mathbf{x}^k), \boldsymbol{\mu} \right) \right) \right] - \mathbb{E}_{\mathbf{x}^b \sim \mathcal{D}_b} \left[ \log \left( 1 - \exp \left( -h \left( D_E^2 \left( g(\mathbf{x}^b), \boldsymbol{\mu} \right) \right) \right) \right) \right], \quad (6)$$

which aims to decrease the Euclidean distances between normal samples $\mathbf{x}^k$ and $\boldsymbol{\mu}$ while increasing the distances for background samples $\mathbf{x}^b$. In Eq. (6), $h(x) = \sqrt{x+1} - 1$, which implies that the Euclidean distance $D_E^2(g(\mathbf{x}), \boldsymbol{\mu})$ is scaled into the range of $(0, 1]$ via $\exp(-h(D_E^2(g(\mathbf{x}), \boldsymbol{\mu})))$.

It is straightforward that the decision rule of Eq. (5) for distance-based classifiers uses the principle of HSCs in a class-wise manner, where the original HSC formulates its decision boundary for anomaly detection as a hypersphere having a constant radius. In other words, the class-wise HSC for the $c$-th class determines whether a test input belongs to the $c$-th class by computing $D_E^2(f(\mathbf{x}), \boldsymbol{\mu}_c)$, where the input is determined as an unknown-class sample if the entire class-wise HSCs reject the data item. Thus, a proper BCR strategy for distance-based classifiers should force each KUC sample $\mathbf{x}^b$ to be rejected by the entire class-wise HSCs (increase $D_E^2(f(\mathbf{x}^b), \boldsymbol{\mu}_i)$ for all $i$). Since it is inefficient to consider the entire class-wise HSCs to regularize $\mathbf{x}^b$ at each iteration, we approximate the process by only taking the *closest* class-wise HSC into account (increase $\min_{i \in \{1, \cdots, C\}} D_E^2(f(\mathbf{x}^b), \boldsymbol{\mu}_i)$).

Although one can adopt Eq. (6) to formulate $\mathcal{L}_{bg}$ for distance-based classifiers, scaling $D_E^2(f(\mathbf{x}), \boldsymbol{\mu}_c)$ into $(0, 1]$ via $\exp(-h(D_E^2(f(\mathbf{x}), \boldsymbol{\mu}_c)))$ can be inappropriate to guarantee sufficient spaces for KKC data and simultaneously move KUC samples far away from the limited space of KKC data, since $\exp(-h(D_E^2(f(\mathbf{x}), \boldsymbol{\mu}_c)))$ rapidly decays near $\boldsymbol{\mu}_c$. To overcome the issue, we propose to design $\mathcal{L}_{bg}$ based on the principle of the probability of inclusion (Jain et al., 2014; Bendale & Boult, 2016).

## 3.3 PROBABILITY OF INCLUSION AND CLASS-INCLUSION LOSS

As we described in Section 2.2, the probability of inclusion builds effective CAP models, since it is designed to rapidly decay near extreme data, i.e., $P_I(\mathbf{x}, c) \approx 1$ in the region that a majority of class-$c$ KKC samples are located. In the following, we describe how to utilize the principle of the probability of inclusion when training distance-based classifiers, and then formulate a loss function for BCR.

Based on pre-trained Softmax classifiers, Openmax (Bendale & Boult, 2016) formulated the probability of inclusion via EVT modeling of latent features at inference time. However, it is intractable to use such EVT-based analyses during the training phase of $f$, since it is inappropriate to limit the space of KKC data by analyzing extreme samples at each training iteration. Thus, we use the underlying assumption of LDA to formulate $P_I(\mathbf{x}, c)$, since we define the training and the inference processes of distance-based classifiers based on the principle of LDA. Under the assumption that each class-$c$

latent feature vector is drawn from a unimodal Gaussian distribution $\mathcal{N}(f(\mathbf{x})|\boldsymbol{\mu}_c, \mathbf{I})$, the Euclidean distance $D_E^2(f(\mathbf{x}), \boldsymbol{\mu}_c)$, a simplified version of the Mahalanobis distance, can be assumed to follow the Chi-square distribution having the degree of freedom $n$. Then, we have

$$P\left(D_E^2(f(\mathbf{x}), \boldsymbol{\mu}_c) = t\right) = \frac{t^{\frac{n}{2}-1}}{2^{\frac{n}{2}} \cdot \Gamma(n/2)} \cdot \exp\left(-\frac{t}{2}\right), \tag{7}$$

where $t \geq 0$, $\Gamma(\cdot)$ is the Gamma function, and $n$ is equivalent to the dimension of $f(\mathbf{x})$.

As previous studies (Jain et al., 2014; Bendale & Boult, 2016; Rudd et al., 2018) formulated the probability of inclusion by computing the CDF of the Weibull distribution, we define our $P_I(\mathbf{x}, c)$ as

$$P_I(\mathbf{x}, c) = 1 - \int_0^{D_E^2(\mathbf{x},c)/2} \frac{t^{\frac{n}{2}-1}}{\Gamma(n/2)} \cdot \exp\left(-t\right) dt = \frac{\Gamma(n/2, D_E^2(f(\mathbf{x}), \boldsymbol{\mu}_c)/2)}{\Gamma(n/2)} \tag{8}$$

by using the CDF of Eq. (7), where $\Gamma(\cdot, \cdot)$ is the upper incomplete Gamma function. It is noteworthy that Eq. (8) can be easily computable via `igammac` function in PyTorch (Paszke et al., 2019).

Based on $\mathcal{D}_t$, $\mathcal{D}_b$, and $P_I(\mathbf{x}, c)$ of Eq. (8), the primary objective of our BCR, which is to make each KUC sample located far away from the closest class-wise HSC, can be achieved by using a loss function $\mathcal{L}_{bg,u} = \mathbb{E}_{\mathbf{x}^b \sim \mathcal{D}_b}[-\log(1 - \max_{i \in \{1,\cdots,C\}} P_I(\mathbf{x}^b, i))]$. To compare $P_I(\mathbf{x}, c)$ and $P_H(\mathbf{x}, c) = \exp(-h(D_E^2(f(\mathbf{x}), \boldsymbol{\mu}_c)))$, which was used in Eq. (6), we plot $P_I(\mathbf{x}, c)$ and $P_H(\mathbf{x}, c)$ in Figure 2 with respect to $||f(\mathbf{x}) - \boldsymbol{\mu}_c||$ by assuming $n = 128$. The figure implies that unlike $P_H(\mathbf{x}, c)$, our $P_I(\mathbf{x}, c)$ can effectively assign space for KKC data and force KUC samples to be located outside the space.

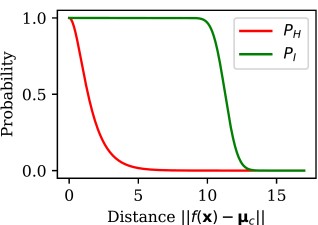

Figure 2: $P_H$ and $P_I$ (Ours)

At training time, note that each $P_I(\mathbf{x}, c) = 0.5$ builds an auxiliary decision boundary between the $c$-th class KKC sample and the others, where $\mathcal{L}_{bg,u}$ ensures a majority of KUC samples located outside the entire class-wise boundaries. However, $\mathcal{L}_{bg,u}$ can be insufficient to achieve robust UUC rejection and closed-set classification results, since it does not control correctly classified KKC samples to be located inside the corresponding class-wise boundaries. Thus, in addition to $\mathcal{L}_{cf}$, we impose a regularization $\mathcal{L}_{bg,k}$ on KKC data to *maintain* closed-set classification performance and enhance the gap between KKC and KUC data in terms of the Euclidean distance. By formulating $\mathcal{L}_{bg,k} = \mathbb{E}_{(\mathbf{x}^k, y) \sim \mathcal{D}_t}[-\mathbb{1}(y = \hat{c}) \log(P_I(\mathbf{x}^k, \hat{c}))]$, where $\hat{c} = \arg\max_{i \in \{1,\cdots,C\}} P_I(\mathbf{x}^k, i)$, we define our BCR loss as $\mathcal{L}_{bg} = \mathcal{L}_{bg,k} + \mathcal{L}_{bg,u}$ and call $\mathcal{L}_{bg}$ the *class-inclusion loss*. In summary, we use

$$\begin{aligned}
\mathcal{L} &= \mathcal{L}_{cf} + \lambda \mathcal{L}_{bg} = \mathcal{L}_{cf} + \lambda(\mathcal{L}_{bg,k} + \mathcal{L}_{bg,u}) \\
&= \mathbb{E}_{(\mathbf{x}^k, y) \sim \mathcal{D}_t}\left[-\log P_d(y|\mathbf{x}^k) - \lambda \mathbb{E}_{\mathbf{x}^b \sim \mathcal{D}_b}\left[\mathbb{1}(y = \hat{c})\log(P_I(\mathbf{x}^k, \hat{c})) + \log(1 - P_I^*(\mathbf{x}^b))\right]\right]
\end{aligned} \tag{9}$$

as our total loss, where $\hat{c} = \arg\max_{i \in \{1,\cdots,C\}} P_I(\mathbf{x}^k, i)$ and $P_I^*(\mathbf{x}^b) = \max_{i \in \{1,\cdots,C\}} P_I(\mathbf{x}^b, i)$.

## 4 EXPERIMENTS

In this section, we compared our class-inclusion loss for distance-based classifiers to the objecto-sphere (Dhamija et al., 2018), the uniformity (also widely known as OE) (Hendrycks et al., 2019), and the energy (Liu et al., 2020) losses for Softmax classifiers. Through our extensive experiments, we aim to show that whether our proposed framework provides competitive UUC rejection results in comparison with the previous BCR methods, while keeping robust closed-set classification results.

### 4.1 EXPERIMENTAL SETTINGS

To assess the OSR performance of each method, we used two experimental settings, which correspond to Sections 4.1.1 and 4.1.2, respectively. When evaluating the OSR performance, we first measured the closed-set classification accuracy. To quantify the performance of UUC data rejection, we also measured the area under the receiver operating characteristic curve (AUROC). Furthermore, we used the open-set classification rate (OSCR) measure by quantifying the correct classification rate at the false positive rate of $10^{-1}$. For in-depth explanation of the OSCR measure, readers are referred to (Dhamija et al., 2018). As $\mathcal{D}_b$, we used ImageNet (Russakovsky et al., 2015), which was also employed in (Li & Vasconcelos, 2020). To ensure that the classes of $\mathcal{D}_b$ and our test sets are disjoint, we used only the remaining classes of the ImageNet dataset, which are not included in the test sets.

Table 1: Comparison with the previous BCR methods in the first setting of our OSR experiments.

| Experiments | Accuracy (↑) | AUROC (↑) | OSCR (↑) |
|---|---|---|---|
| | Objectosphere / Uniformity / Energy / Class-inclusion (Ours) | | |
| SVHN | 0.968 / 0.966 / 0.972 / **0.972** | 0.914 / 0.908 / 0.894 / **0.919** | 0.809 / 0.785 / 0.760 / **0.828** |
| CIFAR10 | 0.964 / 0.964 / 0.956 / **0.973** | 0.942 / 0.923 / 0.933 / **0.946** | 0.851 / 0.814 / 0.807 / **0.869** |
| CIFAR+10 | 0.958 / 0.969 / 0.949 / **0.976** | 0.945 / 0.950 / 0.936 / **0.960** | 0.839 / 0.867 / 0.808 / **0.880** |
| CIFAR+50 | | 0.944 / 0.942 / 0.937 / **0.955** | 0.837 / 0.837 / 0.808 / **0.865** |
| TinyImageNet | 0.778 / 0.779 / 0.715 / **0.801** | 0.755 / 0.771 / 0.727 / **0.784** | 0.484 / 0.488 / 0.357 / **0.492** |

### 4.1.1 OPEN-SET RECOGNITION - SETTING 1

In Setting 1, we split a single dataset into KKCs and UUCs, where we used the KKCs in the training set as $\mathcal{D}_t$, and the KKCs and UUCs in the test set as $\mathcal{D}_{test}^k$ and $\mathcal{D}_{test}^u$, respectively. Following the experiment protocol in (Neal et al., 2018), which were also employed in (Oza & Patel, 2019; Sun et al., 2020), we conducted experiments by using the following standard image datasets: SVHN (Netzer et al., 2011), CIFAR10 & CIFAR100 (Krizhevsky, 2009), and TinyImageNet (Le & Yang, 2015).

**SVHN, CIFAR10** For the SVHN and CIFAR10 datasets, each of which consists of images of 10 classes, each dataset was randomly partitioned into 6 KKCs and 4 UUCs.

**CIFAR+10, CIFAR+50** For the CIFAR+$M$ experiments, we used 4 randomly selected classes of CIFAR10 as KKCs and $M$ randomly selected classes of CIFAR100 as UUCs.

**TinyImageNet** For experiments with a large number of classes, we randomly selected 20 classes of TinyImageNet as KKCs and then used the remaining 180 classes as UUCs.

### 4.1.2 OPEN-SET RECOGNITION - SETTING 2

By using the training and the test sets of a single dataset as $\mathcal{D}_t$ and $\mathcal{D}_{test}^k$, respectively, we employed the test set of another dataset relatively close to $\mathcal{D}_t$ as $\mathcal{D}_{test}^u$ in Setting 2. Adopting the experiment settings in (Yoshihashi et al., 2019) and (Liang et al., 2018), we used the entire classes of a dataset as KKCs for CIFAR10 & CIFAR100. For UUC dataset, TinyImageNet, LSUN (Yu et al., 2015), and iSUN (Xu et al., 2015) were selected. TinyImageNet and LSUN consists of 10,000 test samples each, where the samples in each dataset were resized (RE) or cropped (CR) into the size $32 \times 32$. The iSUN dataset has 8,925 test samples and they were also resized into the size of $32 \times 32$. The modified datasets can be obtained in the Github repository of (Liang et al., 2018).

### 4.2 TRAINING DETAILS

For $f$, we employed the Wide-ResNet (WRN) (Zagoruyko & Komodakis, 2016) and then used its penultimate layer $f(\mathbf{x}) \in \mathbb{R}^n$ for the latent feature vector of each input sample $\mathbf{x}$. For CIFAR10 and TinyImageNet, we used WRN 40-2 with a dropout rate of 0.3, where WRN 28-10 was employed for CIFAR100 with the same dropout rate. For SVHN, we used WRN 16-4 with a dropout rate of 0.4.

For the entire BCR methods, we set the mini-batch sizes of KKC training samples and KUC samples to 128. We kept $\lambda$ as a constant during training, i.e., each $f$ was trained with the BCR method *from scratch*. To select hyperparameters and margin parameters of the previous regularization methods, we followed the official implementations [1,2,3]. For SVHN, CIFAR10, CIFAR100, and TinyImageNet, we trained the corresponding classifiers for 80, 100, 200, and 200 epochs, respectively, where we used the stochastic gradient descent for optimization. For SVHN and the other datasets, we used initial learning rates of 0.01 and 0.1, respectively, and a cosine learning rate decay (Loshchilov & Hutter, 2016). We also used the learning rate warm-up strategy for the first 5 epochs of each training process.

**Average runtime** We conducted all the experiments with PyTorch and a single GeForce RTX 3090 GPU. At each trial in the CIFAR10 experiment of Setting 1, the running time of each training epoch

---

[1] https://github.com/Vastlab/Reducing-Network-Agnostophobia
[2] https://github.com/hendrycks/outlier-exposure
[3] https://github.com/wetliu/energy_ood

Table 2: Comparison with the previous methods in the second setting of our OSR experiments. The corresponding classification accuracy values are reported in the first column.

| $\mathcal{D}_t/\mathcal{D}_{test}^k$ | $\mathcal{D}_{test}^u$ | AUROC (↑) | OSCR (↑) |
| --- | --- | --- | --- |
| | | Objectosphere / Uniformity / Energy / Class-inclusion (Ours) | |
| CIFAR10 0.940 / 0.939 / 0.925 / **0.947** | ImageNet-CR | **0.988** / 0.986 / 0.981 / 0.987 | 0.929 / 0.928 / 0.894 / **0.931** |
| | ImageNet-RE | 0.979 / 0.984 / 0.972 / **0.984** | 0.923 / 0.926 / 0.886 / **0.927** |
| | LSUN-CR | **0.994** / 0.990 / 0.989 / 0.993 | 0.938 / 0.931 / 0.904 / **0.940** |
| | LSUN-RE | 0.985 / 0.988 / 0.984 / **0.990** | 0.928 / 0.931 / 0.897 / **0.935** |
| | iSUN | 0.985 / 0.989 / 0.984 / **0.991** | 0.928 / 0.932 / 0.896 / **0.936** |
| | **Average** | 0.986 / 0.987 / 0.982 / **0.989** | 0.929 / 0.930 / 0.895 / **0.934** |
| CIFAR100 0.727 / 0.735 / 0.705 / **0.779** | ImageNet-CR | 0.886 / 0.929 / **0.925** / 0.922 | 0.641 / 0.686 / 0.652 / **0.693** |
| | ImageNet-RE | 0.815 / 0.910 / **0.934** / 0.902 | 0.572 / 0.674 / 0.658 / **0.683** |
| | LSUN-CR | **0.967** / 0.931 / 0.901 / 0.965 | 0.685 / 0.680 / 0.643 / **0.751** |
| | LSUN-RE | 0.844 / 0.930 / **0.959** / 0.945 | 0.608 / 0.695 / 0.684 / **0.731** |
| | iSUN | 0.842 / 0.923 / **0.954** / 0.943 | 0.603 / 0.689 / 0.680 / **0.729** |
| | **Average** | 0.871 / 0.925 / 0.935 / **0.935** | 0.621 / 0.685 / 0.663 / **0.717** |

took 28 seconds for our method, where its OSR evaluation required approximately 6.5 seconds. We observed that the other methods take similar running time at their training and inference phases.

## 4.3 EXPERIMENT RESULTS

The OSR results of our framework and the previous methods are reported in Tables 1 and 2. All the reported values were averaged over five randomized trials, by randomly sampling seed, split of KKCs and UUCs, and class-wise anchors. In the tables, ↑ indicates that it is better to have larger values for the corresponding measure. Also, an underlined value is marginally worse than the best score (bold).

**Setting 1** For the first setting, Table 1 compares our proposed BCR methods for distance-based classifiers with the previous methods designed for Softmax classifiers. The results demonstrate that our proposed method obtained robust UUC rejection results, which were superior to the results of the previous approaches. It is noteworthy that our method achieved higher classification accuracy values, which were critical in acquiring better OSR results in terms of OSCR, than the previous methods. Such results imply that the proposed framework effectively satisfies the two essential requirements described in Section 2.1, bounding open-space risk and ideally balancing it with empirical risk.

**Setting 2** In Table 2, we present our experiment results of the second setting. When using CIFAR10 and CIFAR100 as KKC datasets, the table shows that our model can achieve the highest closed-set classification accuracy, which is consistent with the experiment results of Setting 1. Furthermore, by averaging the AUROC and the OSCR values over the various UUC datasets, the table shows that our model outperformed the previous methods in UUC sample rejection and OSR.

In summary, the experiment results show that our proposed method can successfully train a robust open-set classifier. To further investigate the effectiveness of our framework in another domain, we compared our class-inclusion loss to the uniformity loss in text classification applications.

**Text classification** For text classification, we used 20 Newsgroups and WikiText103 for KKCs and KUCs, respectively, and trained a simple GRU model (Cho et al., 2014) for $f$ as in (Hendrycks et al., 2019). For UUCs, we used Multi30K, WMT16, and IMDB. Since the margin parameters of the objectosphere and the energy losses selected for image classification were not suitable for the text classification, we tested the uniformity loss for comparison. In Table 3, we present the results, where we additionally reported the area under the precision-recall curve (AUPR) and the false-positive rate at $95\%$ true-positive rate (FPR95). As it outperformed the uniformity loss in image classification tasks, our method also showed significantly better performance in text classification experiments.

## 4.4 ABLATION STUDY AND DISCUSSIONS

For ablation study, we further analyzed our BCR method by using various $\lambda$ and class-wise anchor initialization & updating methods in our loss function. Also, we compared the proposed method to the triplet loss (Schroff et al., 2015) and analyzed each component of our class-inclusion loss. Using the

Table 3: Comparison with the previous BCR method in text classification experiments.

| $\mathcal{D}_t/\mathcal{D}_{test}^k$ | $\mathcal{D}_{test}^u$ | AUROC ($\uparrow$) | AUPR ($\uparrow$) | FPR95 ($\downarrow$) | OSCR ($\uparrow$) |
|---|---|---|---|---|---|
| | | Uniformity / Class-inclusion (Ours) | | | |
| 20 Newsgroups 0.719 / **0.749** | Multi30k | 0.997 / **0.997** | **0.998** / 0.997 | **0.002** / 0.010 | 0.715 / **0.745** |
| | WMT16 | **0.997** / 0.996 | **0.997** / 0.995 | **0.010** / 0.016 | 0.715 / **0.742** |
| | IMDB | 0.805 / **0.999** | 0.692 / **0.999** | 0.367 / **0.003** | 0.585 / **0.747** |
| | **Average** | 0.933 / **0.997** | 0.896 / **0.997** | 0.126 / **0.010** | 0.672 / **0.745** |

CIFAR10 and TinyImageNet experiments in Setting 1, we present quantitative results in Appendix.

**Selecting $\lambda$** Conducting OSR experiments with $\lambda \in \{0.1, 0.5, 1, 5, 10\}$, we observed that $\lambda = 5$ and $\lambda = 1$ showed the best OSR results in the CIFAR10 and TinyImageNet experiments of Setting 1, respectively, where the results are presented in Section A.1. In our additional experiments, $\lambda = 5$ yielded the best results in the SVHN and CIFAR + $M$ experiments of Setting 1 and the CIFAR10 experiments of Setting 2, where $\lambda = 0.5$ showed the best results in the CIFAR100 experiments. Since the other $\lambda$ values did not significantly degrade OSR performance, one can flexibly select $\lambda$, where we empirically observed that a lower $\lambda$ value is better when handling more KKCs.

**Class-wise anchors** In Section 3.1, we mentioned that each class-wise anchor was sampled from the standard Gaussian distribution. Although one can use data-dependent initialization approaches, which are 1) train a Softmax classifier and then use its empirical class mean vectors as class-wise anchors of a distance-based classifier and 2) compute class mean vectors from the initial feature representations of a distance-based classifier and then use the vectors as class-wise anchors, such approaches showed worse results than the random initialization method. In addition, we observed that the strategy of updating or training class-wise anchors was not effective to achieve robust OSR performance. We present the corresponding experiment results and discussions in Section A.2.

**Triplet loss** To the best of our knowledge, we propose the first distance-based BCR method for the OSR problem. Since the triplet loss (Schroff et al., 2015) has been widely employed to control the distances between latent feature vectors effectively, we formulated another distance-based BCR strategy with the triplet loss for comparison. We present in-depth details and experiment results for the regularization method using the triplet loss in Section A.3, where the experiment results demonstrate that our class-inclusion loss is more effective than the triplet loss-based regularization.

**Loss function** Note that our loss function consists of three components $\mathcal{L}_{cf}$, $\mathcal{L}_{bg,k}$, and $\mathcal{L}_{bg,u}$. By using various compositions of the components, we verified in Section A.4 that the three components are necessary to obtain robust closed-classification and UUC rejection results in our BCR method.

**Vanilla distance-based classifiers** To show the effectiveness of our method, we assessed the OSR performance of vanilla distance-based classifiers (trained solely based on $\mathcal{L}_{cf}$), where we present the results in the form of (Accuracy / AUROC / OSCR). For the CIFAR10 and TinyImageNet experiments of Setting 1, we obtained the results of (0.962 / 0.757 / 0.470) and (0.785 / 0.629 / 0.315), respectively. In addition, for the CIFAR10 and CIFAR100 experiments of Setting 2, the OSR results averaged over the five UUC test sets in vanilla distance-based classifiers were (0.936 / 0.838 / 0.709) and (0.766 / 0.807 / 0.549), respectively. Comparing these results to the results in Tables 1 and 2, we show that our regularization strategy can significantly improve the OSR performance of distance-based classifiers.

## 5 CONCLUDING REMARKS

In this paper, we propose a novel BCR method to train open-set classifiers that can provide robust OSR results with a simple inference step. By employing distance-based classifiers with the principle of LDA, we designed a novel class-inclusion loss based on the principle of probability of inclusion, which effectively limits the feature space of KKC data in a class-wise manner and then regularizes UUC samples to be located far away from the limited space. Through our extensive experiments and ablation study, we present that our method can achieve robust UUC rejection performance, while maintaining high closed-set classification accuracy. As this paper aims to improve the reliability of modern DNN-based classifiers, we hope our work to enhance reliability and robustness in various classification applications by providing a novel methodology of handling UUC samples.

## 6 ETHICS STATEMENT

Although there are many beneficial applications of classifiers and anomaly detectors, they may lead to inadvertent discrimination and encoding societal biases in decision-making systems. This is one of the cases where ethical considerations depend strongly on the specific applications.

We use several standard datasets including the ImageNet dataset. Although they have been widely used in research, some of the datasets, created from images available on the web, include images of people, raising ethical questions related to human-derived data, such as encoding of biases and under-representation issues. The creators of ImageNet are also aware of some of these concerns, and have been attempted to address them for the past few years (https://image-net.org/update-sep-17-2019.php).

## 7 REPRODUCIBILITY STATEMENT

We describe details about distance-based classifiers, our loss function, and the corresponding training process in Sections 3.1, 3.3, and 4.2. Also, Section 4.1 provide the list of the datasets, which are used in this paper. Our source code and the corresponding instructions will be released.

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

## A    EXPERIMENT RESULTS OF ABLATION STUDIES

To supplement our discussions in Section 4.4, this section provides quantitative experiment results. Based on the CIFAR10 and TinyImageNet experiments of Setting 1, we present OSR results with various selections of $\lambda$ (Section A.1). Also, we report OSR results by using various class-wise anchor initialization & updating methods (Section A.2). For the settings of fixed and trainable anchors in Section A.2, we compare our class-inclusion loss to the triplet loss (Section A.3). Furthermore, we provide additional OSR results by varying the components of our class-inclusion loss (Sections A.4).

### A.1    HYPERPARAMETER $\lambda$

This subsection presents OSR results by employing various selections of the hyperparameter $\lambda$ in our loss function $\mathcal{L} = \mathcal{L}_{cf} + \lambda\mathcal{L}_{bg}$. In Table 4, we report the OSR experiment results of our method with $\lambda \in \{0.1, 0.5, 1, 5, 10\}$. Since the other $\lambda$ values did not significantly degrade OSR performance, one can flexibly select the parameter $\lambda$ to balance empirical and open-space risks. The experiment results in Sections A.2 and A.3 are also provided by using various $\lambda \in \{0.1, 0.5, 1, 5, 10\}$.

Table 4: OSR experiment results by using various $\lambda$ in our class-inclusion and the triplet losses. In each cell, the results are presented in the form of (Accuracy ($\uparrow$) / AUROC ($\uparrow$) / OSCR ($\uparrow$)).

| Parameter $\lambda$ | CIFAR10 | | TinyImageNet | |
|---|---|---|---|---|
| | Class inclusion | Triplet | Class inclusion | Triplet |
| 0 (Vanilla) | 0.962 / 0.757 / 0.470 | – | 0.785 / 0.629 / 0.306 | – |
| 0.1 | 0.966 / 0.894 / 0.740 | 0.963 / 0.820 / 0.537 | 0.789 / 0.766 / 0.442 | 0.787 / 0.746 / 0.431 |
| 0.5 | 0.968 / 0.927 / 0.810 | 0.968 / 0.842 / 0.572 | 0.793 / 0.775 / 0.463 | 0.785 / 0.729 / 0.426 |
| 1 | 0.973 / 0.934 / 0.839 | 0.966 / 0.860 / 0.628 | 0.801 / 0.784 / 0.492 | 0.793 / 0.714 / 0.423 |
| 5 | 0.973 / 0.946 / 0.869 | 0.965 / 0.872 / 0.628 | 0.795 / 0.784 / 0.479 | 0.785 / 0.707 / 0.360 |
| 10 | 0.967 / 0.946 / 0.862 | 0.958 / 0.856 / 0.597 | 0.785 / 0.698 / 0.358 | 0.738 / 0.637 / 0.220 |

### A.2    CLASS-WISE ANCHORS

This subsection presents OSR results by using various initialization and updating methods of anchors in our approach. In Table 5, we first present OSR results based on *trainable* class-wise anchors. In this setting, we set each entry of $\boldsymbol{\mu}_c$ as a trainable parameter and then optimized the anchors along with our model parameters via the same training objective $\mathcal{L} = \mathcal{L}_{cf} + \lambda\mathcal{L}_{bg}$ and optimization processes.

Table 5: OSR experiment results using trainable class-wise anchors in our loss and the triplet loss. In each cell, the results are presented in the form of (Accuracy ($\uparrow$) / AUROC ($\uparrow$) / OSCR ($\uparrow$)).

| Parameter $\lambda$ | CIFAR10 | | TinyImageNet | |
|---|---|---|---|---|
| | Class inclusion | Triplet | Class inclusion | Triplet |
| 0.1 | 0.966 / 0.894 / 0.740 | 0.964 / 0.767 / 0.318 | 0.795 / 0.443 / 0.038 | 0.798 / 0.383 / 0.052 |
| 0.5 | 0.968 / 0.927 / 0.810 | 0.967 / 0.829 / 0.540 | 0.782 / 0.487 / 0.050 | 0.795 / 0.380 / 0.043 |
| 1 | 0.966 / 0.937 / 0.833 | 0.968 / 0.840 / 0.577 | 0.765 / 0.503 / 0.061 | 0.798 / 0.383 / 0.046 |
| 5 | 0.971 / 0.946 / 0.860 | 0.967 / 0.872 / 0.667 | 0.768 / 0.457 / 0.041 | 0.769 / 0.420 / 0.055 |
| 10 | 0.967 / 0.940 / 0.827 | 0.956 / 0.845 / 0.580 | 0.588 / 0.516 / 0.050 | 0.771 / 0.426 / 0.051 |

Comparing the CIFAR10 experiment results in Tables 4 and 5, we observed that there exist only a negligible difference of OSR performance between our fixed and the trainable anchor strategies. However, the TinyImageNet experiment results in the tables show that the fixed anchor strategy with random initialization can yield more accurate UUC rejection and OSR results.

To further investigate various anchor initialization and updating methods, we additionally assess the OSR performance of our method by employing the following three initialization methods:

**Initial**    Compute anchors from the initial feature representations of a distance-based classifier.
**Transfer**    Train a Softmax classifier and then use its empirical means for a distance-based classifier.
**Random (Ours)**    Randomly sample each class-wise anchor from a standard Gaussian distribution.

For each initialization method, we tested two anchor updating strategies: 1) fix the anchors during our training phase, 2) update the anchors via moving average, which was also used in (Wen et al., 2019).

Table 6: OSR experiment results by using various initialization and updating methods for class-wise anchors. In each cell, the results are shown in the form of (Accuracy ($\uparrow$) / AUROC ($\uparrow$) / OSCR ($\uparrow$)).

| Anchor | $\lambda$ | CIFAR10 | | TinyImageNet | |
|---|---|---|---|---|---|
| | | Fix | Update | Fix | Update |
| Initial | 0.1 | 0.956 / 0.801 / 0.498 | 0.962 / 0.852 / 0.490 | 0.576 / 0.378 / 0.012 | 0.775 / 0.415 / 0.049 |
| | 0.5 | 0.964 / 0.897 / 0.638 | 0.967 / 0.915 / 0.768 | 0.576 / 0.596 / 0.042 | 0.734 / 0.446 / 0.058 |
| | 1 | 0.957 / 0.893 / 0.756 | 0.967 / 0.919 / 0.801 | 0.536 / 0.530 / 0.017 | 0.740 / 0.429 / 0.042 |
| | 5 | 0.958 / 0.913 / 0.727 | 0.961 / 0.918 / 0.742 | 0.506 / 0.426 / 0.007 | 0.726 / 0.377 / 0.027 |
| | 10 | 0.901 / 0.917 / 0.760 | 0.957 / 0.894 / 0.650 | 0.228 / 0.481 / 0.052 | 0.632 / 0.438 / 0.036 |
| Transfer | 0.1 | 0.969 / 0.890 / 0.719 | 0.965 / 0.863 / 0.715 | 0.792 / 0.539 / 0.092 | 0.765 / 0.390 / 0.035 |
| | 0.5 | 0.972 / 0.923 / 0.803 | 0.971 / 0.909 / 0.765 | 0.802 / 0.554 / 0.111 | 0.777 / 0.425 / 0.049 |
| | 1 | 0.972 / 0.938 / 0.838 | 0.971 / 0.924 / 0.808 | 0.796 / 0.544 / 0.093 | 0.780 / 0.392 / 0.054 |
| | 5 | 0.970 / 0.937 / 0.815 | 0.972 / 0.926 / 0.790 | 0.779 / 0.551 / 0.120 | 0.738 / 0.421 / 0.038 |
| | 10 | 0.963 / 0.941 / 0.828 | 0.964 / 0.928 / 0.776 | 0.757 / 0.448 / 0.078 | 0.761 / 0.377 / 0.046 |
| Random | 0.1 | 0.966 / 0.894 / 0.740 | 0.959 / 0.782 / 0.263 | 0.789 / 0.766 / 0.395 | 0.784 / 0.436 / 0.041 |
| | 0.5 | 0.968 / 0.927 / 0.810 | 0.960 / 0.862 / 0.604 | 0.793 / 0.775 / 0.463 | 0.777 / 0.412 / 0.049 |
| | 1 | 0.973 / 0.934 / 0.839 | 0.960 / 0.860 / 0.558 | 0.801 / 0.784 / 0.492 | 0.767 / 0.438 / 0.044 |
| | 5 | 0.973 / 0.946 / 0.869 | 0.953 / 0.778 / 0.262 | 0.795 / 0.784 / 0.479 | 0.738 / 0.409 / 0.042 |
| | 10 | 0.967 / 0.946 / 0.862 | 0.947 / 0.740 / 0.191 | 0.785 / 0.698 / 0.358 | 0.683 / 0.440 / 0.046 |

The results of Tables 5 and 6 present that the OSR results of our approach, which randomly sampled class-wise anchors and then fixed them, are superior to those of the other initialization and updating methods. In the following, we discuss the advantages of random sampling and fixed anchor strategies.

- In contrast to classical closed-set classification, our BCR method for OSR should ensure *sufficient distance gaps* between inter-class latent features (inter-class separability) and force class-wise features to have compact feature space (intra-class compactness) by contrasting KKC samples and KUC data located near the KKC data. As discussed in (Izmailov et al., 2020), the Euclidean distances between anchors drawn from a standard Gaussian distribution are sufficiently large in expectation, which is suitable for our BCR method.
- Our loss is based on the probability of inclusion of Eq. (8), where $P_I = 0.5$ builds a hypersphere decision boundary in a class-wise manner at training time. The class-inclusion loss makes repulsion forces between KUC samples and the corresponding closest anchors, while simultaneously making KKC samples to be located inside the corresponding hypersphere boundary. In this regularization method, it is desirable to fix class-wise anchors to control each KKC or KUC sample in a consistent direction. In other words, the strategy of updating or training class-wise anchors, which also changes the location of class-wise hypersphere boundaries, is likely to obstruct the process of contrasting KKC and KUC samples.

### A.3 Triplet loss-based regularization

This section compares our BCR method to a regularization method based on the triplet loss $\mathcal{L}_{tri}$, in order to show that our class-inclusion loss is an effective BCR method for distance-based classifiers. Following the conventional definition of the triplet loss, we simply formulated $\mathcal{L}_{tri}$ by setting class-wise anchors, KKC training data, and KUC data as anchors, positive samples, and negative samples, respectively. Then, we compared our loss to $\mathcal{L}_{tri}$ in fixed and trainable anchor settings.

In Tables 4 and 5, which presented the OSR results of distance-based classifiers with fixed class means and trainable class means, respectively, we reported experiment results by using the triplet loss as $\mathcal{L}_{bg}$. Since we observed that training classifiers solely based on the triplet loss $\mathcal{L} = \mathcal{L}_{tri}$ yields significantly worse OSR results in comparison with the regularization method $\mathcal{L} = \mathcal{L}_{cf} + \lambda \mathcal{L}_{tri}$, we employed $\mathcal{L}_{tri}$ as a regularization loss function for BCR. The results show that our proposed method (class-inclusion loss) outperforms the regularization method using the triplet loss. Also, the discussions in Sections A.1 and A.2 can also be applicable to the triplet loss-based regularization.

## A.4 Loss function formulation

In our class-inclusion loss $\mathcal{L} = \mathcal{L}_{cf} + \lambda(\mathcal{L}_{bg,k} + \mathcal{L}_{bg,u})$, $\mathcal{L}_{cf}$ is a conventional negative log-likelihood loss for the distance-based classifier of Eq. (4). Based on our probability of inclusion of Eq. (8), we formulated $\mathcal{L}_{bg,k}$ and $\mathcal{L}_{bg,u}$ for regularization. This subsection investigates the necessity of the three terms $\mathcal{L}_{cf}$, $\mathcal{L}_{bg,k}$, and $\mathcal{L}_{bg,u}$, since both of the $\mathcal{L}_{cf}$ and $\mathcal{L}_{bg,k}$ terms aim to force KKC training samples located close to the corresponding class-wise anchors. We employed various compositions of the three terms to formulate loss functions and then reported their OSR results in Table 7.

As our original $\mathcal{L}_{bg,k}$ term only controls KKC training samples having correct class predictions with $\mathcal{L}_{bg,k} = \mathbb{E}_{(\mathbf{x}^k,y)\sim\mathcal{D}_t}[-\mathbb{1}(y=\hat{c})\log(P_I(\mathbf{x}^k,\hat{c}))]$ and $\hat{c} = \arg\max_{i\in\{1,\cdots,C\}} P_I(\mathbf{x}^k,i)$, we define a modified version $\mathcal{L}_{bg,k}^* = \mathbb{E}_{(\mathbf{x}^k,y)\sim\mathcal{D}_t}[-\log(P_I(\mathbf{x}^k,y))]$ to correctly remove $\mathcal{L}_{cf}$ from $\mathcal{L}$.

Table 7: OSR experiment results based on various settings of loss functions.

| Loss function $\mathcal{L}$ | Parameter $\lambda$ | CIFAR10 | TinyImageNet |
|---|---|---|---|
| | | Accuracy (↑) / AUROC (↑) / OSCR (↑) | |
| $\mathcal{L}_{bg,k}^* + \mathcal{L}_{bg,u}$ | – | 0.937 / 0.932 / 0.798 | 0.629 / 0.688 / 0.334 |
| $\mathcal{L}_{cf} + \lambda\mathcal{L}_{bg,u}$ | 0.1 | 0.963 / 0.821 / 0.509 | 0.793 / 0.734 / 0.408 |
| | 0.5 | 0.961 / 0.810 / 0.488 | 0.779 / 0.732 / 0.424 |
| | 1 | 0.958 / 0.806 / 0.485 | 0.787 / 0.726 / 0.392 |
| | 5 | 0.963 / 0.812 / 0.503 | 0.767 / 0.737 / 0.397 |
| | 10 | 0.960 / 0.798 / 0.459 | 0.779 / 0.732 / 0.424 |
| $\mathcal{L}_{cf} + \lambda(\mathcal{L}_{bg,k}^* + \mathcal{L}_{bg,u})$ | 0.1 | 0.965 / 0.868 / 0.674 | 0.795 / 0.781 / 0.483 |
| | 0.5 | 0.972 / 0.930 / 0.822 | 0.800 / 0.763 / 0.461 |
| | 1 | 0.973 / 0.933 / 0.835 | 0.785 / 0.720 / 0.378 |
| | 5 | 0.966 / 0.931 / 0.815 | 0.639 / 0.630 / 0.278 |
| | 10 | 0.956 / 0.919 / 0.771 | 0.458 / 0.629 / 0.178 |
| $\mathcal{L}_{cf} + \lambda(\mathcal{L}_{bg,k} + \mathcal{L}_{bg,u})$ | 0.1 | 0.966 / 0.894 / 0.740 | 0.789 / 0.766 / 0.442 |
| | 0.5 | 0.968 / 0.927 / 0.810 | 0.793 / 0.775 / 0.463 |
| | 1 | 0.973 / 0.934 / 0.839 | 0.801 / 0.784 / 0.492 |
| | 5 | 0.973 / 0.946 / 0.869 | 0.795 / 0.784 / 0.479 |
| | 10 | 0.967 / 0.946 / 0.862 | 0.785 / 0.698 / 0.358 |

In Table 7, it is shown that our loss function ($\mathcal{L}_{cf} + \lambda(\mathcal{L}_{bg,k} + \mathcal{L}_{bg,u})$) can achieve robust OSR results for various selections of $\lambda$. As $\mathcal{L}_{bg,k}^*$ (or $\mathcal{L}_{bg,k}$) only forces KKC samples to be located inside the class-wise hypersphere boundaries of the probability of inclusion, which rapidly decays from 1 to 0, the loss term is not sufficient to achieve high closed-set classification accuracy. On the contrary, KKC samples cannot be forced to be located inside the boundaries when using $\mathcal{L}_{cf}$ without $\mathcal{L}_{bg,k}$, thereby making inaccurate OSR results. In summary, both $\mathcal{L}_{cf}$ and $\mathcal{L}_{bg,k}$ are necessary to conduct accurate classification and increase the distance gap between KKC and KUC data, respectively. It is noteworthy that our formulation (using both $\mathcal{L}_{cf}$ and $\mathcal{L}_{bg,k}$) is beneficial in maintaining robust classification accuracy. Also, our method, which only regularizes KKC samples having correct class prediction, can mitigate overfitting issues in comparison with the approach of ($\mathcal{L}_{cf} + \lambda(\mathcal{L}_{bg,k}^* + \mathcal{L}_{bg,u})$).

## B Additional experiments

In additional experiments, we employed CIFAR10 for $\mathcal{D}_t$. For Setting 1 of our OSR experiments, we split the 10 classes of CIFAR10 into 6 known known classes (KKCs) and 4 unknown unknown classes (UUCs). In Setting 2, we used CIFAR10 and another dataset as KKCs and as UUCs, respectively. For the first setting of our experiments, we report quantitative results in the form of (**Accuracy / AUROC / OSCR**). In this section, we denote Settings 1 and 2 by `S1` and `S2` for simplicity.

### B.1 Quantitative Results with ResNet-18 Architecture

In Section 4, we used the Wide-ResNet (WRN) architectures as feature extractors for the image datasets. To further investigate the effectiveness of our method, we used another standard classifier architecture, ResNet-18 (He et al., 2016). In `S1`, we obtained quantitative results of (**0.963 / 0.945 /**

**0.844**), (**0.966 / 0.950 / 0.845**), (**0.967 / 0.945 / 0.848**), and (**0.971 / 0.947 / 0.850**) for the regularization methods using the objectosphere (Dhamija et al., 2018), the uniformity (Hendrycks et al., 2019), the energy (Liu et al., 2020), and our class-inclusion losses, respectively. Such quantitative OSR results and the results in Table 8 imply that our method can outperform the previous BCR methods with ResNet-18, as we observed in the experiments using the WRN architectures.

Table 8: Comparison with the previous methods in the second setting of our OSR experiments using ResNet-18. The corresponding classification accuracy values are reported in the first column.

| $\mathcal{D}_t/\mathcal{D}^k_{test}$ | $\mathcal{D}^u_{test}$ | AUROC (↑) | OSCR (↑) |
|---|---|---|---|
| | | Objectosphere / Uniformity / Energy / Class-inclusion (Ours) | |
| | ImageNet-CR | 0.982 / 0.979 / 0.983 / **0.987** | 0.932 / 0.928 / 0.917 / **0.941** |
| | ImageNet-RE | 0.977 / 0.982 / 0.975 / **0.988** | 0.918 / 0.934 / 0.909 / **0.945** |
| CIFAR10 | LSUN-CR | 0.991 / 0.984 / 0.987 / **0.993** | 0.934 / 0.935 / 0.919 / **0.946** |
| 0.937 / 0.949 / 0.933 / **0.951** | LSUN-RE | 0.987 / 0.987 / 0.986 / **0.990** | 0.932 / 0.938 / 0.918 / **0.942** |
| | iSUN | 0.987 / 0.986 / 0.987 / **0.994** | 0.932 / 0.938 / 0.919 / **0.946** |
| | **Average** | 0.985 / 0.984 / 0.984 / **0.991** | 0.930 / 0.935 / 0.916 / **0.944** |

## B.2 LATENT FEATURE SPACE VISUALIZATION

By visualizing feature spaces via t-SNE (Van der Maaten & Hinton, 2008), we compared Softmax classifiers and distance-based classifiers, and their regularized versions in S1 and S2. We selected ImageNet-RE for UUC data in S2, since the results of Table 2 imply that the dataset is more difficult to be recognized by open-set classifiers than the others. In each t-SNE result, black dots indicate UUC samples. For the other colors, each distinct color represents the corresponding class of KKCs.

### B.2.1 VANILLA CLASSIFIERS (NO BACKGROUND-CLASS REGULARIZATION)

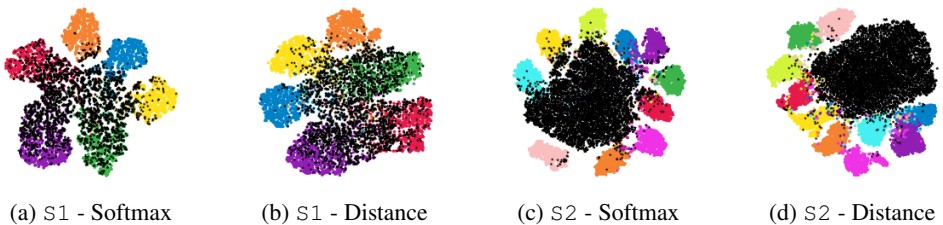

    (a) S1 - Softmax      (b) S1 - Distance      (c) S2 - Softmax      (d) S2 - Distance

Figure 3: t-SNE results of vanilla Softmax classifiers and distance-based classifiers.

Figure 3 depicts the t-SNE results of vanilla Softmax and distance-based classifiers, which are trained only with $\mathcal{L} = \mathcal{L}_{cf}$, in S1 and S2. The visualization results imply that in comparison with the UUC data used in S2, it would be much more difficult to distinguish UUC samples in S1 from KKCs.

### B.2.2 BACKGROUND CLASS REGULARIZED CLASSIFIERS

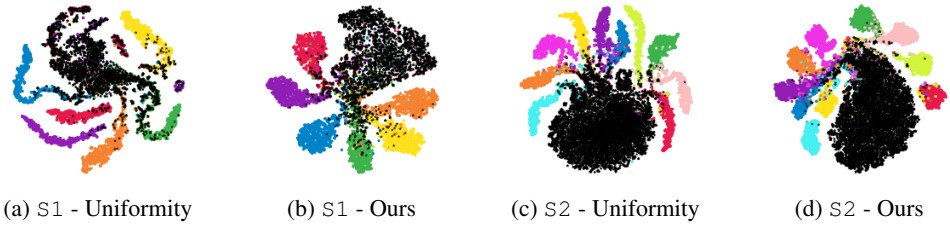

    (a) S1 - Uniformity      (b) S1 - Ours      (c) S2 - Uniformity      (d) S2 - Ours

Figure 4: t-SNE results of regularized Softmax classifiers and distance-based classifiers.

Figure 4 presents the t-SNE results of Softmax and distance-based classifiers trained with BCR ($\mathcal{L} = \mathcal{L}_{cf} + \lambda\mathcal{L}_{bg}$) in S1 and S2. For the Softmax and distance-based classifiers, we employed the uniformity loss (Hendrycks et al., 2019) and our class-inclusion loss as $\mathcal{L}_{bg}$, respectively. In comparison with Figure 3, Figure 4 shows that such regularization techniques assist classifiers to learn robust latent feature representations in distinguishing UUCs from KKCs, especially in S1. Also, as mentioned in Section 2.3, the figure shows that the latent feature space regularized by the uniformity loss can yield inaccurate results in distance-based post-classification analysis. By applying the Openmax approach (Bendale & Boult, 2016) to the model regularized by the uniformity loss in S1 (Figure 4. (a)), we obtained the OSR performance of **(0.923 / 0.886 / 0.752)**, which is worse than the original result **(0.964 / 0.923 / 0.814)** and the result of our framework **(0.973 / 0.946 / 0.869)**.

