# OpenReview forum: "Distance-Based Background Class Regularization for Open-Set Recognition"
_ICLR.cc/2022/Conference — ICLR 2022 Submitted_

### Official Review · Reviewer_rop6 · 2021-10-22

**Correctness:** 3
**Technical Novelty And Significance:** 2
**Empirical Novelty And Significance:** 3
**Recommendation:** 6
**Confidence:** 5

**Main Review:**

In this paper, the authors propose a new method for open set recognition. To this end, the authors use distance based classifiers and they minimize the distance between the class samples and their corresponding means for the labeled data samples and try to enforce the background data samples to be outside the class acceptance regions.
The strengths of the paper can be summarized as follows:
1) The paper is written well and the motivation and the proposed methodology are described well.
2) The authors aim to propose compact acceptance regions for the known classes and they enforce the background samples to lie outside of these regions. The background samples help to return more compact and correct class regions. This idea makes sense and there are recent studies applying the same idea.
3) Experimental results are good in the sense that the proposed method outperforms the competing methods.
The weaknesses of the paper can be summarized as follows:
1) The idea of using distance-based classifiers for returning compact class regions where the known class samples lie inside these regions and negative (background) samples lie far from these regions have been used before. The authors cited some of these studies, but some important references are missing. Among these, the method using center loss function [R1], polyhedral conic classifiers [R2] and Convolutional prototype networks [R3] should be mentioned. Especially, the deep polyhedral conic classifier [R2] implements exactly the same idea. It returns bounded convex acceptance regions for the class samples and enforces the negative samples to lie outside of these regions.
The authors use a similar idea, and they enforce to minimize the distances between the known class samples and their corresponding class centers. They also use background class samples to lie outside the class acceptance regions. To this end, they adopt the idea from Explainable Deep One-Class classification method and they introduce a slightly different loss term that utilized background samples. Therefore, the overall novelty is limited in my opinion.
2) I strongly believe that setting class specific means in the proposed methodology is problematic. The authors randomly select the class means from the standard Gaussian distribution and fix them. This is unacceptable. It should be noted that the features change in each epoch in deep neural network classifiers, and the class means also change in the process. Therefore, the class means must be also updated based on the changed features. This way, local similarities between the semantic classes can be preserved (For example, in the feature space we expect that the class samples belonging to the cat and dog classes lie in close regions and they are probably far from other classes such as aeroplanes, cars, etc.). But when the class samples are chosen randomly and the class samples are enforced to lie in the vicinity of these randomly chosen centers the semantic similarities are completely destroyed. Note that Izmailiov et al. use this setting in the generative models and this is not appropriate for the classification setting. Furthermore, they also recommend the updating class centers.
The authors report that updating class mean vectors yield worse results than fixed class mean vectors. In contrast, both [R1] and [R2] also update the class mean vectors and report good accuracies. The authors must look into this issue in more details (change parameter settings or update centers using the formulations given in [R1] with appropriate weights).
3) The authors use ImageNet dataset for background classes. I was wondering if the overlapping classes between tested datasets and this one are removed. Also, the authors use a completely different setting (using a background class) in their experiments and therefore it is hard to compare the results to the some published related methods. Of course, it is hard to implement every method, but the authors must definitely give results for a distance metric learning method using triplet loss function. Anchors can be set class means and constraints can be easily formed by using known class labels and background class labels. Lastly, an ablation study showing the importance of the utilized background class is necessary in my opinion to judge the effect of background class samples for returning more compact class acceptance regions.
References:
[R1] Wen et al., A comprehensive study on center loss for deep face recognition, Int. Journal of Computer Vision, 127, 2019.
[R2] Cevikalp et al., Deep compact polyhedral conic classifier for open and closed set recognition, Pattern Recognition, 2021.
[R3] Yang et al., Convolutional prototype network for open set recognition IEEE Trans. on Pattern Analyis Machine Intelligence, 2020.


**Summary Of The Paper:**

In this paper, the authors propose a new method for open set recognition. To this end, the authors use distance based classifiers and they minimize the distance between the class samples and their corresponding means for the labeled data samples and try to enforce the background data samples to be outside the class acceptance regions. Then the samples coming from the unknown classes can be rejected based on the distances to the centers of the known classes. The authors use ImageNet data samples are used as background class samples. The proposed method is compared to related open set recognition methods and better accuracies are reported.

**Summary Of The Review:**

In general, the novelty is kind of limited but the experimental results are quite good. There are some issues that must be addressed regarding the setting class centers. Also, more comparison to the related methods and some ablations studies are needed in experiments. A borderline paper closer to the acceptance region in my opinion.

---

> ### Author Response · Authors · 2021-11-23
> **Response to Reviewer rop6 [2/2]**
>
> **[Additional references and our novelty]**
>
> We appreciate the reviewer for introducing additional references related to distance-based classification methods. We added the references in the last paragraph of Section 3.1 of our revised manuscript.
>
> Our main contribution is to design a novel background-class regularization that can effectively utilize background class samples in distance-based classification schemes.
>
> Although the distance-based regularization strategy using background samples was employed in [3,4], the previous works are limited to one-class classification (anomaly detection) tasks. As we observed that the method cannot be directly applied to multi-class classification settings, we carefully designed a novel loss function that can effectively regularize distance-based classifiers with no loss of classification accuracy, which is a very significant point of novelty.
>
> [3] Ruff et al., Rethinking assumptions in deep anomaly detection, arXiv:2006.00339.
>
> [4] Liznerski et al., Explainable deep one-class classification, ICLR21.
>
> **[Classes of background data]**
>
> To ensure that the classes of our background set and test sets are disjoint in our experiments, we used the remaining classes of the ImageNet dataset as our KUCs, which are not included in test data.
>
> **[Additional experiments using triplet loss]**
>
> We appreciate the reviewer for suggesting a baseline method using the triplet loss [5]. Addressing the reviewer’s suggestion, we formulated a triplet loss by following its conventional definition. We used our class-wise anchors, KKC data, and KUC data as anchors, positive samples, and negative samples, respectively, in the loss function.
>
> In Section A.3 of our revised manuscript, we compared our BCR method to a regularization method using the triplet loss L_tri.
>
> In Tables 4 and 5, which presented the OSR results of distance-based classifiers with fixed class means and trainable class means, respectively, we reported experiment results by using the triplet loss as L_bg. Since we observed that training classifiers solely based on the triplet loss (L = L_tri) yields significantly worse OSR results in comparison with the regularization approach (L = L_cf + $\lambda$ L_tri), we employed L_tri as a regularization loss function for BCR.
>
> The results of Tables 4 and 5 show that our proposed method (class-inclusion loss) outperforms the triplet loss-based regularization.
>
> [5] Schroff et al., FaceNet: A unified embedding for face recognition and clustering, CVPR15.
>
> **[Comparison with vanilla distance-based classifiers]**
>
> To address the reviewer’s concern, we added an ablation study about the open-set recognition performance of vanilla distance-based classifiers in the last part of Section 4.4.
>
> The experiment results show that our background-class regularization strategy is effective to formulate more compact class acceptance regions, thus enhancing open-set recognition performance.

---

> ### Author Response · Authors · 2021-11-23
> **Response to Reviewer rop6 [1/2]**
>
> **[Initialization and updating methods of class-wise anchors]**
>
> We agree with the reviewer that our strategy, which randomly sampled class-wise anchors and then fixed them during the training phase, can have weak points when modeling semantic similarity in classical classification problems. However, in contrast to the classical setting which aims to analyze only a closed-set of classes, our method should effectively handle background classes along with the known classes to train a robust open-set classifier. Although it can have disadvantages in modeling the semantic similarity between classes, we believe that our method is suitable for this background class regularization (BCR) setting. Also, it is noteworthy that the empirical results show that our method can achieve robust classification accuracy and sometimes improves the classification performance of vanilla classifiers. In the following, we provide the corresponding discussions and experiment results.
>
> 1.  Our open-set classifier should effectively distinguish known-unknown class (KUC) or unknown-unknown class (UUC) samples (located near known-known class (KKC) data) from KKC samples at training and inference time, respectively. Therefore, it would be desirable to achieve high inter-class separability by ensuring sufficient distance gaps between the class-wise anchors. As discussed in [1], which proposed a generative classifier for semi-supervised classification (not for sample generation), the Euclidean distances between anchors are sufficiently large in expectation if they are drawn from a standard Gaussian distribution, which is suitable for our BCR method.
>
> 2.  When conducting distance-based latent feature regularization, it is reasonable to force KKC samples to be located near the corresponding class-wise anchors while moving KUC samples far away from the anchors. To achieve the objective, our method makes repulsion forces between KUC samples and the corresponding closest anchors while simultaneously forcing KKC samples to be located inside the corresponding hypersphere boundary, which is formulated via the probability of inclusion. To effectively distinguish UUCs from KKCs in this regularization method, it is desirable to force each KKC or KUC feature to move in a consistent direction. In other words, the strategy of updating or training class-wise anchors, which also changes the location of class-wise hypersphere boundaries, is likely to obstruct the process of contrasting KKC and KUC samples.
>
> To address the reviewer’s concern and supplement our claim, we presented OSR results based on various anchor initialization and updating methods in Section A.2 of the revised manuscript.
>
> In our experiments, we employed the following three initialization methods:
>
> 1) Compute anchors by averaging the class-wise initial latent features of a feature extractor, where this method yields inaccurate initial class-wise anchors.
> 2) Successfully train a Softmax classifier and then use its empirical class-wise means as anchors of a distance-based classifier. The semantic similarity between the known classes modeled by the SoftMax classifier can be transferred to the distance-based classifier.
> 3) Randomly sample each class-wise anchor from a standard Gaussian distribution (Ours). This method ensures large distance gaps between the class-wise anchors.
>
> For each initialization method, we tested two anchor updating strategies: 1) fix the anchors during our training phase, 2) update the anchors via moving average, which was also used in [2].
>
> The results of Table 6 present that the OSR results of our approach, which randomly samples class-wise anchors and then fix them, are superior to those of the other anchor initialization and updating methods. The anchor updating strategy was beneficial only when class-wise anchors were inaccurately initialized.
>
> In Table 5, we also presented OSR results by using trainable class-wise anchors, where we set each entry of anchors as a trainable parameter and then optimized the anchors along with our model parameters via the same training objective and optimization processes. The results also show that the fixed anchor strategy with random initialization can achieve more effective UUC rejection and OSR performance in a distance-based BCR strategy.
>
> [1] Izmailov et al., Semi-supervised learning with normalizing flows, ICML20.
>
> [2] Wen et al., A comprehensive study on center loss for deep face recognition, International Journal of Computer Vision, 2019.

---

### Official Review · Reviewer_Mkdh · 2021-10-31

**Correctness:** 3
**Technical Novelty And Significance:** 2
**Empirical Novelty And Significance:** Not applicable
**Recommendation:** 5
**Confidence:** 4

**Main Review:**

The strengths of the paper:
+ The paper is well-written and easy to read.
+ A distance-based background class regularization (BCR) method using the principle of linear discriminant analysis is proposed to address the problem of open-set recognition.
+ Ablation study is well designed to show the performance of the proposed distance-based method.

The weaknesses of the paper:
- The novelty of this work is limited. The distance-based classifier that uses the principle of linear discriminant analysis has been studied in previous method, e.g., [1] Thomas Mensink, Jakob Verbeek, Florent Perronnin, Gabriela Csurka, Distance-Based Image Classification: Generalizing to New Classes at Near-Zero Cost, IEEE Transactions on Pattern Analysis and Machine Intelligence, 2013.
- The experimental evaluation of this paper is not sufficient. The current version lacks comparisons with existing Softmax based methods, e.g.,  [2] Weiyang Liu, Yandong Wen, Zhiding Yu, Ming Li, Bhiksha Raj, and Le Song, Sphereface: Deep hypersphere embedding for face recognition, IEEE Conference on Computer Vision and Pattern Recognition, 2017; [3] Feng Wang, Jian Cheng, Weiyang Liu, and Haijun Liu, Additive margin softmax for face verification, IEEE Signal Processing Letters, 2018.

**Summary Of The Paper:**

This paper introduces a distance-based background class regularization (BCR) method for open-set recognition (OSR), in which the distance-based classifier that uses the principle of linear discriminant analysis is utilized to limit the feature space of known-class data in a class-wise manner and make background-class samples far away from the limited feature space. Experiments show the robust OSR performance of the proposed method.

**Summary Of The Review:**

The paper is well-written and easy to read. However, the novelty is limited and the experiments are not sufficient.

---

> ### Author Response · Authors · 2021-11-23
> **Response to Reviewer Mkdh**
>
> **[Novelty and insufficient evaluation]**
>
> We appreciate the reviewer for introducing previous distance-based classification methods and related Softmax-based classifiers. However, our novelty is not the distance-based classifier itself, as we already mentioned the references of the nearest class mean classifier [1] and the prototypical classifier [2] in our original manuscript. Our main contribution is to design a novel background class regularization method that can effectively utilize background-class samples in the distance-based classification approaches.
>
> To the best of our knowledge, we are the first to discuss the necessity of distance-based background class regularization methods for open-set recognition (OSR) and propose a reasonable solution. Although Dhamija et al. [3] proposed a background class regularization (BCR) method for OSR, which requires an additional data-dependent margin parameter, it cannot bound the open-space risk and has limited OSR performance. Our method overcomes such problems by proposing a simple yet effective BCR method for distance-based classifiers.
>
> In designing a loss function, we observed that the previous distance-based regularization [4,5] for one-class classification (anomaly detection) tasks cannot be directly applied to multi-class classification settings. Thus, we carefully designed a novel loss function that can effectively regularize distance-based classifiers with no loss of classification accuracy, which is a very significant point of our novelty.
>
> Therefore, we strongly believe that our method should be compared to previous BCR methods, which employ known-unknown class (KUC) samples at training time to conduct effective OSR via simple inference processes, and our experiments are sufficient to show the effectiveness of the proposed method.  To address the reviewer's concern about insufficient evaluation, we compared our method to another distance-based regularization method using the triplet loss [6] to supplement our experiments, where the results are presented in Sections 4.4 and A.3 of the revised manuscript. Since the triplet loss has been a popular method to control the distances between latent feature vectors, we formulated the loss by using our class-wise anchors, known-known class (KKC) data, and KUC data as anchors, positive samples, and negative samples, respectively. Our experiment results show that our method using the proposed class-inclusion loss is more effective than the regularization method based on the triplet loss.
>
> [1] Mensink et al., Metric learning for large scale image classification: Generalizing to new classes at near-zero cost, ECCV12.
>
> [2] Snell et al., Prototypical networks for few-shot learning, NIPS17.
>
> [3] Dhamija et al., Reducing network agnostophobia, NIPS18.
>
> [4] Ruff et al., Rethinking assumptions in deep anomaly detection, arXiv:2006.00339.
>
> [5] Liznerski et al., Explainable deep one-class classification, ICLR21.
>
> [6] Schroff et al., FaceNet: A unified embedding for face recognition and clustering, CVPR15.

---

> > ### Comment · Reviewer_Mkdh · 2021-11-30
> > **Thank you for the rebuttal**
> >
> > The rebuttal clarifies some of the comments, however, I still keep the original rate due to the marginal novelty.

---

### Official Review · Reviewer_ujMG · 2021-11-02

**Correctness:** 3
**Technical Novelty And Significance:** 2
**Empirical Novelty And Significance:** 2
**Recommendation:** 5
**Confidence:** 3

**Main Review:**

Strengths:

The main contribution is a loss function that includes background class regularization (BCR) based on probability of inclusion estimated by the CDF of Weibull distribution, though BCR and CDF of Weibull distribution are not new. Empirical results indicate that the proposed method compares favorably against a few existing algorithms on a few datasets.  The paper is generally well written.

Weakness:

Since both L_cf and L_bg,k try to get known instances close to their class means,  an ablation study on excluding either one of them would be interesting.  Also, a more detailed analysis and discussion of why L_bg,k is needed would be helpful.

Minor:

Paragraph next to Figure 2: Eq 7 -> Eq 8 for P_I

**Summary Of The Paper:**

For open set recognition (OSR), the authors propose using background data to represent instances from unknown classes.  The background data are from data outside the known classes.  Based on LDA, they proposed using the distance from an instance to the class mean to estimate the posterior class probability via softmax.  The class means are randomly chosen beforehand.  Their loss functions includes cross entropy, L_cf, and background class regularization (BCR), which is denoted as L_bg.   For L_bg, they introduce probability of inclusion, P_I, which is estimated by the CDF for Weibull distribution.  With the background data, they calculate L_bg,u  based on 1-P_I.  With the known data, they calculate L_bg,k based on P_I.

Empirical evaluations have two settings.  The unknown classes from the first setting are from the same dataset as the known classes.  In the second setting the unknown classes are from another dataset.  Background data are from another dataset.  In the both settings, the proposed method compares favorably against a few existing algorithms on a few datasets.



**Summary Of The Review:**

While the empirical results are favorable, justification and analysis of including L_bg,k could be more detailed.

---

> ### Author Response · Authors · 2021-11-23
> **Response to Reviewer ujMG**
>
> **[Our formulation of the probability of inclusion]**
>
> Previous post-classification methods formulated the probability of inclusion by analyzing latent feature vectors of a pre-trained Softmax classifier via the Weibull distribution and then limited the feature spaces of known-known class (KKC) samples with compact boundaries. However, it is inappropriate to limit the space of KKCs by analyzing latent feature vectors during the training phase. It is noteworthy that we utilized the class-wise Gaussian assumption of distance-based classifiers to make a novel formulation of the probability of inclusion based on the CDF of Chi-squared distribution, which is suitable for regularizing the classifiers.
>
>
> **[Ablation study on L_cf and L_bg,k]**
>
> Based on our own formulation of the probability of inclusion, we proposed L_bg,k and L_bg,u for regularization. To address the reviewer’s concern, we provided an ablation study and the corresponding discussion about the two loss terms in Sections 4.4 and A.4.
>
> Since our original L_bg,k term only controls KKC training samples having correct class predictions, we define a modified version L_bg,k^* (presented in Section A.4) to correctly replace L_cf.
>
> In Table 7 of our revised manuscript, it is shown that our loss function can achieve robust OSR results for various selections of $\lambda$. As L_bg,k^*  (or L_bg,k) only aims to force KKC samples to be located inside the class-wise hypersphere boundaries of the probability of inclusion, which rapidly decays from 1 to 0, the loss term is not sufficient to achieve high closed-set classification accuracy. On the contrary, it is insufficient to distinguish known-unknown class (KUC) samples from KKC samples when using L_cf without L_bg,k, since KKC samples cannot be forced to be located inside the boundaries.
>
> In summary, both L_cf and L_bg,k are necessary to conduct accurate classification and increase the distance gap between KKC and KUC data, respectively, and it is noteworthy that our formulation (using both L_cf and L_bg,k) is beneficial for achieving robust open-set recognition (OSR) results while maintaining high classification accuracy. Also, our method, which only regularizes KKC samples having correct class prediction, can mitigate overfitting issues in comparison with the loss function of L_cf + $\lambda$ ( L_bg,k^*+L_bg,u).
>
>
> **[Minor comment]**
>
> In our revised manuscript, we revised the minor issue that the reviewer pointed out.

---

> > ### Comment · Reviewer_ujMG · 2021-11-25
> > **Comment on reviewer response**
> >
> > 1. Thanks for the ablation study to show having both L_cf and L_bg,k is better than having only either one.
> >
> > "we define a modified version L_bg,k^* (presented in Section A.4) to correctly replace L_cf."
> >
> > Not sure why L_bg,k^* correctly "replaces" L_cf.   Maybe you mean similar to L_cf, all instances from one class are included, instead of only the correctly classified instances are included.   Then I would expect L_bg,k^* to perform better than L_bg,k because the wrongly classified instances are also be included. However, that is not case in Table 7 in A.4.   Any thoughts?
> >
> > 2. Thanks for the added explanation for the need of  L_cf and L_bg,k.
> >
> > In A.4: "As L∗bg,k (or Lbg,k) only forces KKC samples to be located inside the
> > class-wise hypersphere boundaries of the probability of inclusion, which rapidly decays from 1 to 0,
> > the loss term is not sufficient to achieve high closed-set classification accuracy."
> >
> > Not sure why it is not sufficient.
> >
> > In A.4: "On the contrary, KKC samples cannot be forced to be located inside the boundaries when using L_cf without L_bg,k"
> >
> > Although samples might not be forced inside the boundaries, L_cf will try to move the samples close to the anchors.  More insights would be nice.

---

> > > ### Author Response · Authors · 2021-11-25
> > > **Additional explanations**
> > >
> > > We appreciate the reviewer for raising insightful questions and sorry for our insufficient explanation in the previous response. As the reviewer pointed out, we defined L_bg,k^* to formulate L = L_bg,k^* + L_bg,u, since L_bg,k only controls the correctly classified instances.
> > >
> > > For additional explanation, we introduce the following two types of decision boundaries used in our method.
> > >
> > > **Type 1)** Class decision boundaries between known-known classes (KKCs), which are based on the principle of linear discriminant analysis (Equation 4). L_cf makes KKC training samples to be correctly classified by the boundaries.
> > >
> > > **Type 2)** Class-wise boundaries designed *only for* background-class regularization, which are formulated by the probability of inclusion (Equation 8). L_bg,k makes correctly classified KKC samples located inside the corresponding class-wise boundaries.
> > >
> > > Based on the two boundaries, we provide additional explanations to address the reviewer's questions:
> > >
> > > __In A.4: "On the contrary, KKC samples cannot be forced to be located inside the boundaries when using L_cf without L_bg,k". More insights would be nice.__
> > >
> > > Consider a binary classification problem, where we have class-wise anchors $\mu_1$ and $\mu_2$. Then, a decision boundary of Type 1 should be formulated between $\mu_1$ and $\mu_2$. Also, R1 and R2, hypersphere-shaped decision boundaries of Type 2, are formulated around $\mu_1$ and $\mu_2$, respectively.
> > > As we mentioned, the objective of L_cf can be satisfied if $||{\bf x} - \mu_1|| < ||{\bf x} - \mu_2||$ for a class 1 sample ${\bf x}$. However, this objective does not necessarily make the entire class 1 samples to be located inside R1 ($||{\bf x} - \mu_1|| < r_1$, where $r_1$ is the radius of R1) at training time .
> > >
> > > Since L_bg,u forecs known-unknown class (KKC) samples to be located outside R1 and R2 by using the probabilities of inclusion, which rapidly decays near R1 and R2, we believe that it is insufficient to effectively contrast KKC and KUC samples without L_bg,k. Our experiments with L = L_cf + $\lambda$ L_bg,u support our claim by showing worse UUC rejection and OSR performance. Also, it is noteworthy that L_bg,k is likely to keep correctly classified KKC samples located near the corresponding class-wise anchors when moving KUC samples far away from the anchors, thus maintaining high classification accuracy.
> > >
> > > __In A.4: "As L_bg,k^* (or L_bg,k) only forces KKC samples to be located inside the class-wise hypersphere boundaries of the probability of inclusion, which rapidly decays from 1 to 0, the loss term is not sufficient to achieve high closed-set classification accuracy." Not sure why it is not sufficient.__
> > >
> > > We designed L_bg,k and L_bg,u only for regularizing a distance-based classifier using L_cf. The principle of our background class regularization is to reserve *sufficient* latent feature spaces for KKCs and ensures KKC and KUC samples to be located inside and outside the reserved spaces.
> > > As we reserve the spaces with *sufficiently large* hypersphere boundaries at training time, there may exist overlapping regions between the class-wise boundaries .
> > >
> > > Recall the binary classification problem. Assume that there exist an overlapping region between the hypersphere-shaped boundaries R1 and R2 and a class 1 sample ${\bf x}$ is located inside the region. Although ${\bf x}$ is in the intersection of R1 and R2, it does not ensure correct classification, i.e., ${\bf x}$ can be determined as class 2 by a decision boundary of Type 1 (incorrectly classified).
> > >
> > > However, the objective of L_bg,k^* is already satisfied (the probability of inclusion $\approx 1$ inside the union of R1 and R2) in such case. Thus, unlike L_cf, L_bg,k^* is insufficient to ensure accurate classification results. Our experiments with L = L_bg^* + L_bg,u support our claim by showing worse closed-set classification accuracy.
> > >
> > > __Then I would expect L_bg,k^* to perform better than L_bg,k because the wrongly classified instances are also be included. However, that is not case in Table 7 in A.4. Any thoughts?__
> > >
> > > As the reviewer pointed out, it seems reasonable to replace L_bg,k to L_bg,k^* in our loss function. However, we believe that controlling the entire KKC training data via the additional regularization of L_bg,k^* tends to make the classifier converges fast to a suboptimal point and degrade its robustness against $\lambda$.
> > >
> > > Unlike our original formulation L = L_cf + $\lambda$ (L_bg,k + L_bg,u), we observed that L = L_cf + $\lambda$ (L_bg,k^* + L_bg,u) yields worse OSR results especially when using a high $\lambda$. Thus, we used L_bg,k to apply additional regularization to KKC samples only if the samples are correctly classified and then observed that our method provides robust OSR results for various $\lambda$ values.
> > >
> > > To address the reviewer's concern, we will add the corresponding in-depth discussions in the final version of our paper.

---

> > > > ### Comment · Reviewer_ujMG · 2021-12-01
> > > > **Comments on author response**
> > > >
> > > > Thanks for the response.
> > > >
> > > > --- On L_cf being not sufficient:
> > > >
> > > > Eq 5 uses the closest anchor and squared distance within Tau for inferring KKC or unknown.  However, the relationship between sqrt(Tau) and r1 (or r2) is not apparent.    Also sqrt(tau) is the same for any KKC, but r1 and r2 could be different for KKCs.  Since L_cf minimizes the distance of a KKC sample to its anchor, there is an "implicit radius" that contains samples of a KKC.  Hence, understanding the relationship between the "implicit radius" [from L_cf] and r1 (r2) [from L_bg,u] might be important to justify the inclusion or exclusion of L_bg,k.
> > > >
> > > > ---  On L_bg,k being not sufficient:
> > > >
> > > > If the hyperspheres with radii r1 and r2 "overlap", I can see why L_bg,k is not sufficient for classifying the KKCs.  Including evidence to support overlapping hyperspheres would be nice.  Also, it seems if the hyperspheres do not overlap, L_bg,k could be sufficient.
> > > >
> > > > --- On L_bg,k^* vs L_bg,k:
> > > >
> > > > "L_bg,k^* tends to make the classifier converges fast to a suboptimal point" -- this is what the empirical results indicate, why does  L_bg,k^* reach a less optimal point than L_bg,k?

---

> > > > > ### Author Response · Authors · 2021-12-02
> > > > > **Response to the reviewer's comments [2/2]**
> > > > >
> > > > > **On L_bg,k being not sufficient**
> > > > >
> > > > > In our setting where class-wise anchors are independently sampled from the standard Gaussian distribution, the Euclidean distance between two class-wise anchors is $\sqrt{2D}$ in expectation ($D$ is the dimension of latent feature vectors).
> > > > >
> > > > > For instance, the Euclidean distance between two class-wise anchors is $16$ in expectation when we use $128$-dimensional feature vectors of WideResNet 40-2. In this case, Figure 2 in our paper, which plots $P_I$ when $D=128$, implies that there can exist overlapping regions between class-wise boundaries (e.g., $P_I=0.5$). We checked that such overlapping can occur for high-dimensional feature vectors (e.g., $D > 1000$).
> > > > >
> > > > > __On L_bg,k^* vs L_bg,k__
> > > > >
> > > > > We designed our L_bg,k due to the limited empirical OSR performance of L_bg,k^*, especially when $\lambda$ is large. In the training phase, we also observed that the regularization term tends to obstruct the training process of a classifier at early epochs. In the following, we describe our underlying intuitions when designing L_bg,k to resolve the issue.
> > > > >
> > > > > Since BCR methods employ KUC samples relatively close to KKC samples for effective regularization, loss functions for KKCs (or KUCs) tend to simultaneously affect the features of both KKCs and KUCs. For instance, L_bg,u can make some of KKC samples be located far from anchors.
> > > > >
> > > > > Recall the binary classification problem in our previous response.
> > > > > At an early stage of training, assume that a class-1 sample ${\bf x}$ is not correctly classified yet and there exist a KUC sample ${\bf x}^b$ relatively close to ${\bf x}$.
> > > > >
> > > > > In our scenario, we only used L_cf and L_bg,u for ${\bf x}$ and ${\bf x}^b$ before ${\bf x}$ is correctly classified. Then, even when $\lambda$ is large, ${\bf x}$ and ${\bf x}^b$ can be successfully moved into the class-1 region determined by the class decision boundary of Type 1 while keeping sufficiently large distances from $\mu_1$ and $\mu_2$, since L_cf does not strictly limit class-wise feature spaces and L_cf can be decreased by achieving $||{\bf x} - \mu_1|| < ||{\bf x} - \mu_2||$.
> > > > > Then, L_bg,k term is applied to ${\bf x}$ so that a classifier can learn to effectively separate ${\bf x}$ and ${\bf x}^b$ while keeping the correct class prediction of ${\bf x}$.
> > > > >
> > > > > On the other hand, consider a case that an additional regularization L_bg,k^* is applied to ${\bf x}$ along with L_bg,u and $\lambda$ is large.
> > > > > Since L_bg,k^* and L_bg,u, which dominate the entire loss term L, can simultaneously affect ${\bf x}$ even when ${\bf x}$ is not correctly classified yet, the training strategy using L_bg,k^* can make a classifier to find a sub-optimal solution which has lower classification accuracy.
> > > > >
> > > > > ---
> > > > > Thank you again for your insightful comments, and we will reflect the discussions in the final version of our paper.
> > > > >
> > > > > Also, please let us know if any issues remain.
> > > > >
> > > > > Best regards,
> > > > >
> > > > > Authors

---

> > > > > ### Author Response · Authors · 2021-12-02
> > > > > **Response to the reviewer's comments [1/2]**
> > > > >
> > > > > We really appreciate the reviewer for providing follow-up comments.
> > > > >
> > > > > ---
> > > > >
> > > > > **On L_cf being not sufficient**
> > > > >
> > > > > Considering the feature space of a trained classifier, the $\tau$ value, which is a threshold for distinguishing UUC samples from KKC test data, can be flexibly determined by users at inference time. Thus, open-set recognition (OSR) or out-of-distribution (OOD) detection studies usually report their performances without using a specific $\tau$ value, e.g., AUROC or AUPR.
> > > > >
> > > > > In such problem settings, the objective of background class regularization (BCR) is to design a training strategy that can effectively contrast KKC samples to KUC ones in terms of a specific measure (in our case, Euclidean distance) so that one can achieve robust OSR or OOD detection test results by thresholding the measure with an arbitrary $\tau$. In our method, the radius of class-wise boundaries ($r$), which was carefully designed for distance-based regularization, was only employed as an effective tool to increase the distance gaps between KKC and KUC samples at training time. The radius $r$ is not necessarily related to $\tau$, which is a threshold value for inference.
> > > > >
> > > > > Previous background data regularization methods used a similar principle. For instance, Hendrycks et al. [1] forced KUC samples to have high entropy to conduct OOD detection by imposing an arbitrary threshold on maximum softmax probability scores. Also, Liu et al. [2] employed two auxiliary data-dependent margin parameters $m_i$ and $m_o$ to make the “energy” score values of KKC samples and KUC samples close to $m_i$ and $m_o$, respectively ($m_i$ and $m_o$ are only used for regularization and not employed at inference time).
> > > > >
> > > > > Furthermore, as we use a single $\tau$ for any KKC at inference time, we made auxiliary class-wise boundaries for regularization to have the same radius $r$ ($r=r_1=r_2=\cdots$) by computing the probability of inclusion based on the Euclidean distance between a feature vector and a class-wise anchor. It would be a nice future research direction to use a different distance measure for each class.
> > > > >
> > > > > In addition to the $\tau$ issue, the reviewer also pointed out that L_cf can formulate an implicit radius. Although we agree with the reviewer, we believe that it is hard to estimate the implicit radius during (or before) training time and use the radius to formulate L_bg,u (even when training a classifier without regularization). Also, as BCR methods use KUC samples relatively close to KKC samples, L_bg,u can also affect some of KKC samples so that it is more difficult to estimate an implicit radius. Also, such effects of L_bg,u on KKC samples imply that it is desirable to use an additional regularization that can keep KKC samples to be located in a proper feature space. One alternative solution that can contrast KKCs and KUCs by considering an implicit radius can be a triplet loss-based regularization (Reviewer rop6 suggested the method for additional comparison), since the triplet loss can force the distance between a KUC sample and a class-wise anchor to be larger than that between a KKC sample and the anchor. We empirically observed that our strategy, which explicitly defined auxiliary class-wise boundaries by using a function that rapidly decays near the boundaries, shows significantly better OSR results.
> > > > >
> > > > > It is noteworthy that our novel formulation of the probability of inclusion yields effective explicit class-wise boundaries, whose radius are automatically selected with respect to the feature dimension (do not require any data-dependent hyperparameter selection such as energy margin parameters $m_i$ and $m_o$).
> > > > >
> > > > > [1] Hendrycks et al., Deep Anomaly Detection with Outlier Exposure, ICLR 19.
> > > > >
> > > > > [2] Liu et al., Energy-based Out-of-distribution Detection, NeurIPS 20.

---

### Official Review · Reviewer_1wRX · 2021-11-02

**Correctness:** 4
**Technical Novelty And Significance:** 3
**Empirical Novelty And Significance:** 3
**Recommendation:** 8
**Confidence:** 4

**Main Review:**

The main goal of the work is to define a novel open-set classifier on the top of DNN which can improve the detection of unknown classes while keeping the accuracy of original closed-set recognition. This is a well-studied problem and typical applications of image and text classification are considered. A group of different techniques from prior work are combined in a novel way and each part of the model is justified and the whole model is experimentally verified against state-of-the-art.

Strong points

The paper is presenting an interesting approach to open-set recognition in the case of background class information (i.e., data of known-unknown classes) is available. Although, proposed method combines several different techniques (i.e., LDA classifier, inclusion probability modelling, and loss function extended with class-inclusion loss), the model choices are clearly and well justified. Furthermore, the limitations, issues, and differences of the most similar previous works are described in detail to support and motivate the proposed methodologies and model. The claims of model properties are experimentally validated against three previous approaches, and it is shown that open-set recognition rate is improved, and the similar or better close-set accuracy is achieved simultaneously. It is also stated that code and experiments will be available to reproduce the results.

Summary of pros
- Clearly written
- Clear and theoretically sound techniques
- Comprehensive experiments on benchmark datasets
- Reproducibility based on detailed equations and code/experiments to be released

Weak points

Although the whole classifier is well-validated, there is room for improvement in studying the behavior of different parts and parameters of the model. Especially, in ablation study section, to get better view how different part of the model and parameters are behaving, it would be useful to give more detailed description and illustrations: 1) When experimenting with different lambda parameter, it would be useful to show graphically how the change of lambda affects the classification accuracy and OSCR in different datasets. 2) When experimenting different class mean vectors initialization, it would be useful to show the detailed results and collect them to a single table, to get better view how robust they are and how significant the differences are.

As claimed, a simple inference process, loss function with fewer data-dependent parameters, and ability to bound open-space risk without additional post-analysis, are proposed. To strengthen the presentation and claims, it would be good to show how computational complexity and efficiency of the proposed classifier in relation to training and inference times compares to previous methods utilizing SoftMax classifiers.

Summary of cons
- Ablation study is somewhat vague; more detailed and illustrative experiments would strengthen the analysis of model parameters and choices (e.g., selecting lambda, detailed performance of different initialization of class mean vectors)
- Missing computation complexity and efficiency analysis of proposed method in relation to training and inference times (if there are any significant differences compared to previous methods?)

Other (minor) comments and questions
- Eq. 9: It would be useful to write out the final loss equation with classification and class-inclusion parts included. Now they are given in the middle of the text only
- In 4.1: use similar term S1/S2 or setting 1 / setting 2 in each following subsection when referring to different experiments
- Background dataset (ImageNet) could be introduced already in 4.1 (because it is used in both S1 and S2)
- Result presentation (Table 1 and Table 2): Equally good results of previous methods should be also bolded (Table 1: SVHN accuracy of Energy model, Table 2: CIFAR100 average AUROC of Energy model), or are these rounded to third decimal and class-inclusion model is marginally better?
- In ablation studies related to classification accuracy (with conventional loss function), it would be useful to collect these results to a new table for comparison
- How is the best/optimal lambda selected in experiments? How much does it affect the classification accuracy and OSCR numerically?
- How much does different class mean vector initialization strategies (empirical mean, feature mean, random sampling, re-computed empirical mean) affect the model performance? and are there any explanations of why they behave/perform differently?
- Are there some good practices of how “known unknown classes” (KUC) data should be chosen or generated? Could this be automated? How robust will the proposed model be given the different choices of KUC data?

**Summary Of The Paper:**

This paper proposes a new approach for generalized open-set recognition (OSR) in classification setting where the idea is to build a model which could simultaneously detect unknown classes and maintain the accuracy of traditional closed-set classification of known classes. The proposed OSR is based on background class regularization (BCR) where data of known unknown classes are utilized in the training phase of the classifiers based on typical deep neural network (DNN) architectures.

Previous works applying BCR have treated known-classes as one group and unknowns as other with SoftMax classification layer, leading to limited OSR performance, especially near the known classes, whereas proposed work regularizes and limits feature space in class-wise manner, making the data of unknown classes to be located far away from the known classes. Distance-based classifier applying linear discriminative analysis is used to model the latent features from DNN and to design novel loss function which can regulate the solution between known classes and unknown background classes in OSR setting, simultaneously keeping the robust performance of closed-set classification.

Proposed approach is experimentally compared to three previous works applying BCR-based OSR with SoftMax classifiers. Proposed method outperforms those three related methods in well-known public benchmarks datasets of image and text classification tasks. Different performance metrics of classification accuracy, area under the receiver operating characteristic curve (AUROC), and the open-set classification rate (OSCR) are utilized in comparison.

To summarize the contribution, paper presents LDA principled distance-based classifier (applied to output of neural network latent feature layer) and new loss function to train OSR-sensitive classifiers, developed and formulated based on the limitations of previous work and experimentally validated against previous BCR-based approaches.

**Summary Of The Review:**

The paper presents an interesting and novel approach combining several techniques to improve DNN-based open-set recognition with distance-based classifier. Paper is well-written and the methodological choices are properly justified. More detailed ablation study and the analysis of the model efficiency would still improve the presentation and verify the claims better. This is an important and practical problem and can bring new insight for the field. I recommend the paper to be accepted.

---

> ### Author Response · Authors · 2021-11-23
> **Response to Reviewer 1wRX [2/2]**
>
> **[Computational efficiency (average runtime)]**
>
> We observed that the average runtime values of our training and inference processes are similar to those of the previous regularization methods. Specifically, when using a single NVIDIA GeForce RTX 3090 GPU in the PyTorch framework, a single training epoch of our method in the CIFAR 10 experiment of Setting 1 took approximately 28 seconds, where the corresponding OSR inference required approximately 6.5 seconds. Addressing the reviewer’s suggestion, we described the average runtime of our method in Section 4.2 of our revised manuscript.
>
> **[Known-unknown class data selection]**
>
> In Hendrycks et al. [2], it was discussed that the diversity of known-unknown class (KUC) data is an important factor when selecting a dataset for regularization. Furthermore, the authors found that KUC data close to known-known class (KKC) test data can be more effective than those close to unknown-unknown class (UUC) test data. Based on the findings, Li and Vasconcelos [3] improved the approach of [2] by introducing a background data resampling method. The resampling method aims to automatically select effective samples from a large-scale dataset of diverse images based on an assumption that it is sufficient to use a smaller background set that can effectively limit the space of KKCs. Through experiments, the authors verified that the resampling method can improve anomaly detection performance by automatically selecting effective KUC samples located near KKC data.
>
> As reported in [2] and [3], we observed that a large-scale and diverse KUC dataset effectively regularizes classifiers and generalizes to unseen UUC data, where the classifiers showed robust UUC detection performance for various UUC data. For automatic selection or generation of an effective KUC dataset, one can use the resampling method of [3] after collecting large-scale and diverse KUC samples.
>
> [2] Hendrycks et al., Deep anomaly detection with outlier exposure, ICLR19.
>
> [3] Li and Vasconcelos, Background data resampling for outlier-aware classification, CVPR20.
>
> **[Writing Issues]**
>
> Thank you for the detailed comments to improve our manuscript. We revised all the following issues and additional minor problems that the reviewer pointed out.
>
> 1.  In Equation 9, we presented the final loss equation with classification and class-inclusion parts included.
> 2.  We changed the terms S1 and S2 to Settings 1 and 2, and then moved their descriptions to Sections 4.1.1 and 4.1.2, respectively.
> 3.  We moved the description of our background dataset to Section 4.1.
> 4.  In Tables 1 and 2, the scores without boldface are marginally worse than the corresponding best results. To clarify the best scores, we underlined such marginally worse results in the tables.

---

> > ### Comment · Reviewer_1wRX · 2021-11-27
> > **Comments on revised version and response**
> >
> > I would like to thank authors for the response and all the efforts revising the manuscript.
> >
> > 1. More detailed ablation studies (on selecting lambda, and class mean vectors / class-wise anchors initialization and updating strategies) give experimental evidence and justifications of claims that proposed regularization strategy has its advantages, at least on these particular benchmark datasets.
> > 2. A short analysis of average training and inference runtimes shows us computational efficiency of proposed approach as being comparable to previous work (although the metrics of other methods are not shown) and is quite fast to train and to make inferences.
> > 3. My other comments and questions are also clarified and revised to improve the presentation of the paper.
> >
> > Empirical results, especially after the revised ablation study, give evidence that the proposed methodology works in practice and is successfully compared against several related approaches. One future development goal could be the more throughout theoretical analysis of class-wise anchor point behavior in relation to loss function regularization and its parts.

---

> ### Author Response · Authors · 2021-11-23
> **Response to Reviewer 1wRX [1/2]**
>
> **[Ablation studies on $\lambda$ and class-wise anchors]**
>
> Addressing the reviewer’s comments, we added open-set recognition (OSR) experiment results (Table 4) by using various selections of the $\lambda$ parameter in Section A.1. The experiment results show that one can flexibly select the parameter $\lambda$ to balance empirical and open-space risks, and we empirically observed that a lower $\lambda$ value is better when handling more known known classes (KKCs).
>
> Furthermore, we provided additional experiment results (Tables 5 and 6) and the corresponding discussions in Section A.2 of the revised manuscript to support our explanation about anchor initialization and updating methods in Section 4.4. We observed that our strategy, which randomly sampled class-wise anchors and then fixed them, shows the best OSR performance in our experiments. We summarize the corresponding discussions as follows:
>
> 1.  Our open-set classifier should effectively distinguish known-unknown class (KUC) and unknown-unknown class (UUC) samples (located near KKC data) from KKC samples at training and inference time, respectively. Therefore, it would be desirable to achieve high inter-class separability by ensuring sufficient distance gaps between the class-wise anchors. As discussed in [1], which proposed a generative classifier for semi-supervised classification (not for sample generation), the Euclidean distances between anchors are sufficiently large in expectation if they are drawn from a standard Gaussian distribution, which is suitable for our background class regularization (BCR) method.
>
> 2.  When conducting distance-based latent feature regularization, it is reasonable to force KKC samples to be located near the corresponding class-wise anchors while moving KUC samples far away from the anchors. To achieve the objective, our method makes repulsion forces between KUC samples and the corresponding closest anchors while simultaneously forcing KKC samples to be located inside the corresponding hypersphere boundary, which is formulated via the probability of inclusion. To effectively distinguish UUCs from KKCs in this regularization method, it is desirable to force each KKC or KUC feature to move in a consistent direction. In other words, the strategy of updating or training class-wise anchors, which also changes the location of class-wise hypersphere boundaries, is likely to obstruct the process of contrasting KKC and KUC samples.
>
> [1] Izmailov et al., Semi-supervised learning with normalizing flows, ICML20.

---

### Author Response · Authors · 2021-11-23
**General Response**

We would like to thank the reviewers for providing high-quality reviews and constructive feedback. In our general response, we denote Reviewers 1wRX, ujMG, Mkdh, and rop6 as R1, R2, R3, and R4, respectively.

We addressed the reviewers’ concerns and suggestions in our reviewer-specific responses. Based on the valuable suggestions and reviews of the reviewers, we revised our manuscript, where the modifications in the revised version are summarized below:

**[General]**
-   Moved the text classification experiments to Section 4.3.
-   Modified the term “class mean vectors” to “class-wise anchors” in our method.

**[Reviewer-specific]**
-   [R1] Added “Average runtime” in Section 4.2 to discuss the computational efficiency of our method.
-   [R1] Modified Equation 9 to clearly represent the entire loss function.
-   [R1] Modified $\texttt{S1}$ and $\texttt{S2}$ to Settings 1 and 2, respectively.
-   [R1] Moved the explanation of our background dataset to Section 4.1.
-   [R1] Underlined the scores marginally worse than the best scores (bold) in Tables 1 and 2.
-   [R1] Added experiment results about the $\lambda$ parameter in Section A.1.
-   [R1, R4] Added analyses and discussions about anchor initialization and updating methods in Section A.2.
-   [R2] Added experiment results that can further analyze each term of our loss function in Section A.4.
-   [R3, R4] Added more references related to distance-based classification schemes in the last paragraph of Section 2.1 and then clarified our contribution.
-   [R3, R4] Added a comparison between our method and triplet loss-based methods in Section A.3.
-   [R4] Added the open-set recognition results of vanilla distance-based classifiers in the last part of Section 4.4.

---

### Decision · Program_Chairs · 2022-01-20

**Decision:**

Reject

**Comment:**

This paper tackles the open-set recognition problem, specifically the subset that looks at rejecting test data that with unknown classes that are related to the training data. The proposed approach uses an existing distance-based classifier (based on LDA) combined with a new background class regularizer. Results, comparing to a few prior OSR methods, are shown across image/text datasets.

  The reviewers gave a mixed set of scores, with concerns about visualization/ablation studies and the lambda parameter with affect on classification accuracy (1wRX, ujMG, rop6), computation complexity and efficiency (1wRX), limited novelty and discussion of relationship to prior works (Mkdh, rop6), and limited comparison to state of art as only a few algorithms are compared to the proposed approach (Mkdh), and initialization method. Notably, the authors make a strong claim for the latter point that the method should only be compared to previous BCR methods (as opposed to softmax-based classifiers, for example); this seems to ignore whole classes of different methods that can approach the OSR problem. While it is true that comparing to previous BCR methods can directly show your approach is superior to them under similar class of algorithms (thereby showing that it is an improvement), putting the method within the context of the entire literature is absolutely necessary to discuss relative impact to the field. For example, the improvements in AUROC are not that great (and in some cases worse) than even the methods you compare to, while OSCR is improved significantly, so it is not clear how it stacks up with respect to the current state of art. Even if it doesn't beat it, you could argue your contribution, but not presenting it all prevents a holistic perspective that is necessary.

  The authors provided thorough rebuttals, including additional ablations and experiments. However, after the review period the scores remain mixed (5,5,6,8) and the reviewers expressed remaining concerns about novelty and comparison to the current state of art (not just BCR-based methods). As a result of these remaining concerns, I recommend rejection at this time.